# Hydrodynamics of a relativistic charged fluid in the presence of a periodically modulated chemical potential

N. Chagnet[1⋆], K. Schalm[1]

[1] Instituut-Lorentz for Theoretical Physics, Δ-ITP, Leiden University, The Netherlands.
⋆ chagnet@lorentz.leidenuniv.nl

March 30, 2023

## Abstract

We study charged hydrodynamics in a periodic lattice background. Fluctuations are Bloch waves rather than single momentum Fourier modes. At boundaries of the unit cell where hydrodynamic fluctuations are formally degenerate with their Umklapped copy, level repulsion occurs. Novel mode mixings between charge, sound, and their Umklapped copies appear at finite chemical potential — both at zero and finite momentum. We provide explicit examples for an ionic lattice, i.e. a periodic external chemical potential, and verify our results with numerical computations in fluid-gravity duality.

# 1  Introduction

In considering the quantum mechanical wave function of a single electron in a lattice of atoms Bloch had the insight that one should expand the wavefunction in a manner consistent with the discrete periodicity[1]

$$\Psi(x) = \int_{-\frac{\pi}{L}}^{\frac{\pi}{L}} \mathrm{d}k\, e^{ikx} u_k(x) = \sum_n \int_{-\frac{\pi}{L}}^{\frac{\pi}{L}} \mathrm{d}k\, e^{i(k+\frac{2n\pi}{L})x} u_n(k)\ ,$$
$$u_k(x+L) = u_k(x). \tag{1}$$

The novel part of Bloch was its application to quantum wavefunctions rather than waves in general. How waves propagate in periodic structures was already considered by Newton, and that waves in periodic structures exhibit peculiar interference phenomena that we now know as level repulsion/Umklapp/gap opening at Brillouin zone boundaries or Bragg reflection from point-like lattices was already recognized by Kelvin in the 1880s [1]. In electrical engineering the propagation of electromagnetic waves in periodic structures was [2], and is an important topic, see e.g. [3].[2] Also sound waves in lattices were considered from the earliest days up to today, see e.g. [4].

Sound waves, however, are hydrodynamic fluctuations – a long-time long-wavelength perturbation around thermodynamic equilibrium of a conserved charge associated to a global symmetry – and in that sense differ from electromagnetic waves or single particle wavefunctions in that the fundamental equations of motion, i.e. the hydrodynamic conservation laws, are non-linear. The wave-like fluctuations propagate on a background that is itself a full (equilibrium) solution to the non-linear set of equations, and through the non-linearity the properties of the fluctuating waves depend on this background solution. Though gradients are energetically disfavored, through external forcing the equilibrium background can be imprinted with a spatially varying temperature $T(x)$, pressure $P(x)$, or chemical potential $\mu(x)$. Due to the non-linear coupling between fluctuations and the background in hydrodynamics, the wave propagation properties can be self-consistently determined from the (spatially varying) background. This was elucidated particularly clearly in recent years in the context of electron hydrodynamics in systems with random charge impurities [5,6]. Such charge disorder is encoded in a spatially varying chemical potential with average $\mathbb{E}[\mu(x)] = \mu_0$ and variance $\mathbb{E}[\mu(x)\mu(y)] - \mathbb{E}[\mu(x)]\mathbb{E}[\mu(y)] = \sigma_\mu^2 \delta(x-y)$. Quantum mechanical single particle electron motion in the presence of random impurities is a classic condensed matter problem. As Anderson showed, the random wavefunction interference is essentially uniformly destructive; at low temperatures all motion is inhibited and the system becomes an insulator. In the hydrodynamic regime, however, i.e., in

---

[1]The Fourier transform here is chosen with a different convention than the traditional physics convention $f(x) = \int \frac{\mathrm{d}k}{2\pi} \hat{f}(k) e^{ikx}$. This prevents a proliferation of $2\pi$-factors in non-linear terms in dynamical fluctuation equations.

[2]In the latter context Bloch's theorem is known as Floquet's theorem. This is not to be confused with periodically driven Floquet systems, though the underlying mathematics of periodic structures is the same after switching "space" and "time".

a situation where many electrons collectivize to a classical fluid rather than a quantum mechanical wave, the conductivity rather strikingly remains finite indicating the existence of an "incoherent metal" state [6]. Observing this electron hydrodynamics in sufficiently pure 2D systems is currently actively pursued, see e.g. [7] or [8], references therein and the recent review [9].

Here we study not hydrodynamics with random spatial disorder but with strictly periodic modulations of the background, i.e. a lattice. Moreover, we also consider hydrodynamics of a charged rather than a neutral fluid with an eye towards condensed matter systems. Compared to the many existing studies on sound waves in periodic structures, the presence of electromagnetic charge as an additional conserved quantum number changes the fluctuating wave response fundamentally. This is again due to the non-linear nature of the hydrodynamic equations. At finite chemical potential sound mixes with charge diffusion. In a companion article we focus on the significant consequences of this cross-coupling of Bloch modes in a lattice for the measurable DC and AC conductivities in condensed matter systems where this hydrodynamics approach may apply [10]. In this article we provide the deeper hydrodynamic analysis of the full fluctuation spectrum of charged hydrodynamics in a periodic background.

## 2  Hydrodynamics: Set-up and brief review of homogeneous fluctuations

The principal reason that linearized hydrodynamic fluctuations in a lattice background should also be expanded in Bloch modes has already been emphasized: the essence is wave propagation in a periodic structure. Waves are described by coupled first order differential equations of the form [3]

$$(\partial_t + M(x))\phi(x) = 0 \ . \tag{3}$$

If $M(x)$ is periodic $M(x + \frac{2\pi}{G}) = M(x)$, then $\phi(x)$ can be decomposed in Bloch waves[4] $\phi(x) = \sum_n \int_{-G/2}^{G/2} dk\, \phi_n(k) e^{i(k+nG)x}$. Taking $M(x) = -M_0 \partial_x^2 + A\cos(Gx)$ as canonical example, one can solve Eq. (3) perturbatively in $A$. Diagonalizing $M$ in terms of $\phi_n(k) = \sum_p A^p \phi_n^{(p)}(k)/p!$, the lowest eigenvector to first order in $A$ is

$$\phi_n(k) = \phi_n^{(0)}(k) - \frac{A}{2G(G-2k)M_0}\phi_{n-1}^{(0)}(k) - \frac{A}{2G(G+2k)M_0}\phi_{n+1}^{(0)}(k) + \dots \tag{5}$$

---

[3]The standard wave equation $(\partial_t^2 - M_{12}M_{21})\phi_1 = 0$ follows from

$$\begin{pmatrix} \partial_t & M_{12} \\ M_{21} & \partial_t \end{pmatrix} \begin{pmatrix} \phi_1 \\ \phi_2 \end{pmatrix} = 0 \ . \tag{2}$$

[4]The Bloch theorem essentially states that the plane wave decomposition $\phi(x) = \int_{-\infty}^{\infty} dq\, \phi(q) e^{iqx}$ can be segmented into unit cells $q_n \in [(n-1/2)G, (n+1/2)G]$ where $n \in \mathbb{Z}$ labels each cell – or Brillouin zone – as $\phi(x) = \sum_n \int_{(n-\frac{1}{2})G}^{(n+\frac{1}{2})G} dq_n\, \phi(q_n) e^{iq_n x}$. The wavevector in each Brillouin zone can be shifted $q_n = k + nG$ with $k \in [-G/2, G/2]$ such that

$$\phi(x) = \sum_n \int_{-G/2}^{G/2} dk\, \phi(k+nG) e^{i(k+nG)x} \equiv \sum_n \int_{-G/2}^{G/2} dk\, \phi_n(k) e^{i(k+nG)x} \ . \tag{4}$$

The advantage of this decomposition is that discrete periodic shifts $x \to x + 2m\pi/G$ relate modes in different Brillouin zones at the same Bloch momentum $k \in \{-G/2, G/2\}$.

in terms of the unperturbed eigenmodes. This mixing between the different Bloch waves is Umklapp. In this article we shall only focus on these perturbative solutions for small lattice amplitudes.

We also already noted that what is special about hydrodynamics is that the fluctuation equations are themselves a linearization expansion of the fundamental non-linear equations. The principle behind the theory of hydrodynamics is local equilibrium and encoded in the local conservation laws of macroscopic charges, i.e., of a slowly spatially varying energy-momentum tensor $T_{\mu\nu}(x)$ and in the presence of a $U(1)$ charge, a current $J^\mu(x)$. In turn this implies that one can also describe fluid behavior in the presence of a slowly spatially varying external potential whether temperature $T(x)$, pressure $P(x)$, or chemical potential $\mu(x)$.

For simplicity — as well as for the experimental supposition that strongly correlated condensed matter systems can have an emergent Lorentz symmetry at low energies — we shall use $d = 2$ relativistic charged hydrodynamics in this article. In principle all we state also applies to arbitrary $d$ non-relativistic charged hydrodynamics, even if the precise expressions may be subtly different. In relativistic charged hydrodynamics the dynamical equations are simply the conservation equation of the energy-momentum tensor and the charge-current

$$\partial_\mu T^{\mu\nu} = F^{\nu\rho}_{\text{ext}} J_\rho \ , \qquad \partial_\mu J^\mu = 0 \ . \tag{6}$$

Here we have allowed for an external electromagnetic field strength $F^{\mu\nu}_{\text{ext}} = \partial^\mu A^\nu_{\text{ext}} - \partial^\nu A^\mu_{\text{ext}}$ in terms of a local external vector potential. In this paper, we will be interested in taking $A_{\mu,\text{ext}} = (\mu_{\text{ext}}(x), 0, 0)$ with $\mu_{\text{ext}}(x)$ a periodic function. Though again, in principle our results also hold for a spatially varying (external) pressure (see e.g. [11]), or a spatially varying (external) temperature.[5]

The dynamical variables of the fluid are the temperature $T$, the unit timelike velocity vector $u^\mu = (1, v^i)/\sqrt{1 - v^2}$, and the chemical potential $\mu$. Away from equilibrium, the conserved currents in our theory – which we assumed to be parity-invariant, see [12] for more general cases – are given by the constitutive relations at first order in gradients in Landau frame

$$T^{\mu\nu} = \epsilon u^\mu u^\nu + P \Delta^{\mu\nu} - \eta \Delta^{\mu\rho} \Delta^{\nu\sigma} \left( \partial_\rho u_\sigma + \partial_\sigma u_\rho \right) - \Delta^{\mu\nu} \left( \zeta - 2\eta/d \right) \partial_\rho u^\rho \ , \tag{7a}$$

$$J^\mu = n u^\mu - \sigma_Q \Delta^{\mu\nu} \left[ T \partial_\nu \left( \mu/T \right) - F_{\nu\rho,\text{ext}} u^\rho \right] \ . \tag{7b}$$

Here $d = 2$ is the number of spatial dimensions and the shear viscosity $\eta$, the bulk viscosity $\zeta$, and the microscopic conductivity $\sigma_Q$ are hydrodynamic transport coefficients — in principle set by the microscopic details of a given theory, see e.g. [13, 14], in practice phenomenologically determined. $\Delta^{\mu\nu} = \eta^{\mu\nu} + u^\mu u^\nu$ is a projector orthogonal to the fluid velocity. The Landau frame choice is such that at any order in gradients, we have $J^t = n$ and $T^{tt} = \epsilon$.

The above constitutive relations also hold in a static equilibrium background. In a system with Galilean or relativistic Lorentz boost invariance — which we use in this paper — it is convenient to choose the reference frame for which the equilibrium fluid is at rest. In absence of contact to a spatially varying heat bath, the temperature must then also be constant and independent of position. In the presence of a spatially varying external chemical potential $\mu_{\text{ext}}(x)$, the equilibrium solution to the hydrodynamic equations

---

[5]A spatially varying temperature without forcing by contact with a spatially varying heatbath is difficult to have in a static equilibrium configuration, however.

Eqs. ([6]) is then parametrized as

$$v^i = 0 \ , \quad T(t,x) = \bar{T} = T_0 \ , \quad \mu(t,x) = \bar{\mu}(x) = \mu_{\text{ext}}(x) \ ,$$
$$n(t,x) = \bar{n}(x) \ , \quad \epsilon(t,x) = \bar{\epsilon}(x) \ , \quad P(t,x) = \bar{P}(x). \tag{8}$$

In the grand canonical ensemble, the hydrostatic equilibrium yields moreover

$$\nabla_x \bar{P} = \bar{n} \nabla_x \bar{\mu} \ . \tag{9}$$

Throughout this paper we will use the bar notation $\bar{X}$ to denote such static background quantities. For a homogeneous background, they will be spatially constant and we will use a subscript 0 as $X_0$ to denote them. For spatially varying quantities, we will use superscripts $Y^{(n)}$ to describe higher order (Bloch wave) moments

$$Y(x) = \sum_n \int_{-G/2}^{G/2} \mathrm{d}k \, Y^{(n)}(k) e^{i(k+nG)x} \ . \tag{10}$$

The hydrodynamic equations need to be supplemented by an equilibrium equation of state relating the energy density $\epsilon$, the pressure $P$ and the charge density $n$ to solve in terms of the equilibrium values of $T$ and $\mu$. In this paper, we will be considering a general fluid whose equation of state $P(T,\mu)$ determines the thermodynamic equilibrium of the theory. In Sec. [4], we will specialize to conformal systems.

On top of this background, we now consider perturbations $X(t,x) = \bar{X}(x) + \delta X(t,x)$. The conservation equations ([6]) then take the form

$$\partial_t \delta\epsilon + \sigma_Q (\nabla_x \bar{\mu})^2 \delta\lambda_\epsilon = -\sigma_Q (\nabla_x \bar{\mu}) \left[ \nabla_x \delta\lambda_n + \delta E_x \right] - \nabla_x (\bar{\chi}_{\pi\pi} \delta v_x) + (\nabla_x \bar{P}) \delta v_x \ , \tag{11a}$$

$$\partial_t \delta n - \sigma_Q \nabla_x^2 \delta\lambda_n = \sigma_Q \nabla_x \left[ \delta\lambda_\epsilon \nabla_x \bar{\mu} + \delta E_x \right] - \nabla_x (\bar{n} \delta v_x), \tag{11b}$$

$$\partial_t \delta\pi_x - \hat{\eta} \nabla_x^2 \delta v_x = (\nabla_x \bar{\mu}) \delta n - \nabla_x (\bar{n} \delta\lambda_n) - \nabla_x (\bar{\chi}_{\pi\pi} \delta\lambda_\epsilon) - \bar{n} \delta E_x \ , \tag{11c}$$

$$\partial_t \delta\pi_y - \eta \nabla_x^2 \delta v_y = 0 \ , \tag{11d}$$

which can further be simplified into

$$\partial_t \delta\epsilon + \sigma_Q (\nabla_x \bar{\mu})^2 \delta\lambda_\epsilon = -\sigma_Q (\nabla_x \bar{\mu}) \delta E_x^{\text{tot}} - \bar{\chi}_{\pi\pi} \nabla_x \delta v_x - (\nabla_x \bar{\epsilon}) \delta v_x \ , \tag{12a}$$

$$\partial_t \delta n - \sigma_Q \nabla_x \delta E_x^{\text{tot}} = \sigma_Q \nabla_x (\delta\lambda_\epsilon \nabla_x \bar{\mu}) - \nabla_x (\bar{n} \delta v_x), \tag{12b}$$

$$\partial_t \delta\pi_x - \hat{\eta} \nabla_x^2 \delta v_x = -\bar{n} \delta E_x^{\text{tot}} - \nabla_x (\bar{\chi}_{\pi\pi} \delta\lambda_\epsilon) + (\nabla_x \bar{\epsilon}) \delta\lambda_\epsilon \ , \tag{12c}$$

$$\partial_t \delta\pi_y - \eta \nabla_x^2 \delta v_y = 0 \ . \tag{12d}$$

In the previous expression, we have defined a renormalized viscosity $\hat{\eta} \equiv \zeta + \frac{2(d-1)}{d} \eta$. We also introduced the "potential"-variations $\delta\lambda_\epsilon \equiv \frac{\delta T}{T_0}$ and $\delta\lambda_n \equiv \delta\mu - \frac{\bar{\mu}}{T_0} \delta T$ conjugate to the energy and charge densities. The velocity perturbations $\delta v_i$ are conjugate to the momenta $\delta\pi_i$. These are not independent due to the hydrodynamic local equilibrium condition. The momenta $\delta\pi_i$ are related to the velocity perturbations $\delta v_i$ through the constitutive relations $\delta\pi_i = \delta T^{ti} = \bar{\chi}_{\pi\pi} \delta v_i$ at this order.[6] Similarly, the charge and energy densities $\delta n, \delta\epsilon$

---

[6] Formally the susceptibility $\bar{\chi}_{\pi\pi}(x_1, t_2; x_2, t_2) = \frac{\partial}{\partial v_i(x_1, t_1)} \frac{\partial}{\partial v_i(x_2, t_2)} Z(v_i)$ denotes how a local charge density $\pi^i(x_1, t_1)$ is influenced by a (chemical) potential $v_i(x_2, t_2)$ at a different space-time point. Here $Z(v_i)$ is the partition function in the presence of a chemical potential (velocity) $v_i$ for the charge density (momentum) $\pi^i$. In the hydrodynamic limit, however, one assumes that all equilibrium ($t_1 + t_2 = 0$) static ($t_1 - t_2 \to \infty$) charges depend only *locally* on the potentials $\pi^i(v_i(x))$. In a homogeneous equilibrium background where $\bar{\chi}_{\pi\pi}^{\text{static}} = \chi^{\text{static}}(x_1 - x_2)$ this is equivalent to approximating the static susceptibility with its constant part $\bar{\chi}_{\pi\pi}^{\text{static}}(x_1 - x_2) = \chi_{\pi\pi,0} + (x_1 - x_2) \partial_x \bar{\chi}_{\pi\pi}(0) = \chi_{\pi\pi,0} + \dots$. We discuss this in more detail below and in Appendix [A].

are related to the sources $\delta\lambda_n, \delta\lambda_\epsilon$ through the thermodynamic susceptibilities derived in Appendix A. Using that $\bar{\chi}_{\pi\pi} = \bar{\epsilon} + \bar{P}$, we can use the fundamental thermodynamic relation $\bar{\epsilon} + \bar{P} = T_0 \bar{s} + \bar{\mu}\bar{n}$ and the first law of thermodynamics to relate

$$\delta P = \frac{\bar{\chi}_{\pi\pi} - \bar{\mu}\bar{n}}{T_0}\delta T + \bar{n}\delta\mu = \bar{\chi}_{\pi\pi}\delta\lambda_\epsilon + \bar{n}\delta\lambda_n \ . \tag{13}$$

Finally, we introduced an external electric field $\delta E_x \equiv \partial_t \delta A_{x,\text{ext}}$ and in (12) we introduced the total electric field $\delta E_x^{\text{tot}} \equiv \delta E_x + \nabla_x \delta\lambda_n$. In what follows, we will be interested in the hydrodynamics response of the modes $\{\delta\epsilon, \delta n, \delta\pi_x, \delta\pi_y\}$ obeying the equations (12). In the following sections, we will use the static susceptibilities relating the potentials $\{\delta\lambda_\epsilon, \delta\lambda_n, \delta v_x, \delta v_y\}$ to the densities $\{\delta\epsilon, \delta n, \delta\pi_x, \delta\pi_y\}$ to express the equations in terms of the latter only, i.e., we work in the microcanonical ensemble.

## 2.1  Hydrodynamic fluctuations in a homogeneous background

In this section, we first review the hydrodynamics of a long wavelength perturbation above a homogeneous conformal charged fluid. Further details can be found in [15,16]. Since the background is homogeneous, this means every barred quantity will be a constant $\bar{X} = X_0$. The equations (12) decouple into the longitudinal and transverse sectors. We will start by looking at the latter whose equation of motion is simpler. Choosing the wavenumber $k_x$ along the $x$ direction without loss of generality, the transverse fluctuations $\delta\pi_y$ obey

$$\partial_t \delta\pi_y(t,k) + D_\perp k^2 \delta\pi_y(t,k) = 0 \ , \quad D_\perp \equiv \eta/\chi_{\pi\pi,0} \ . \tag{14}$$

This is a simple diffusion equation with the shear diffusion constant $D_\perp$. We can now use the (Fourier-)Laplace transform[7] such that the transverse equation of motion becomes

$$\left(-i\omega + D_\perp k^2\right)\delta\hat{\pi}_y(\omega,k) = \delta\pi_y(t=0,k) = \chi_{\pi\pi,0}\delta v_y(t=0,k) \ . \tag{15}$$

The solution is formally given in terms of the retarded correlator for the transverse momentum which is defined as

$$\delta\pi_y(t,k) = \int_{-\infty}^{\infty} \mathrm{d}t' \, G^R_{\pi_y\pi_y}(t-t',k)\delta v_y(t',k) \ . \tag{16}$$

Using that $\hat{G}^R_{\pi_y\pi_y}(\omega=0,k) = \chi_{\pi\pi,0}$ in the hydrodynamic long wavelength limit (i.e., we only keep the leading term in an expansion in $k$; see footnote 6.), we have

$$\hat{G}^R_{\pi_y\pi_y}(\omega,k) = \chi_{\pi\pi,0}\frac{D_\perp k^2}{D_\perp k^2 - i\omega} \ . \tag{17}$$

The correlator exhibits a pole on the imaginary axis at $\omega = -iD_\perp k^2$ indicative of a purely diffusive mode.

We can now carry the same analysis in the longitudinal sector where the dynamical equations are coupled. Denoting $\delta\phi_a(t,k) = (\delta\epsilon(t,k), \delta n(t,k), \delta\pi_x(t,k))$, the dynamical equations can be written succinctly as

$$\partial_t \delta\phi_a(t,k) + M_{ab}(k)\delta\phi_b(t,k) = 0 \ . \tag{18}$$

---

[7]The Laplace transform is required to have a well-defined right-hand side to our linearized equations. One could also just use a Fourier transform while setting external sources. More details can be found in [15,16].

We can once again use a (Fourier)-Laplace transform to rewrite this system of equations as

$$\hat{K}(\omega, k) \cdot \delta\phi(\omega, k) = \delta\phi(t = 0, k) \tag{19}$$

with the dynamical matrix

$$\hat{K}(\omega, k) \equiv -i\omega \mathbb{1}_3 + M(k) = \begin{pmatrix} -i\omega & 0 & ik \\ -\frac{\chi_{n\epsilon,0}}{d_\chi}\sigma_Q k^2 & \frac{\chi_{\epsilon\epsilon,0}}{d_\chi}\sigma_Q k^2 - i\omega & ik\frac{n_0}{\chi_{\pi\pi,0}} \\ ik\frac{\chi_{nn,0}\chi_{\pi\pi,0} - n_0\chi_{n\epsilon,0}}{d_\chi} & ik\frac{n_0\chi_{\epsilon\epsilon,0} - \chi_{n\epsilon,0}\chi_{\pi\pi,0}}{d_\chi} & \frac{\hat{\eta}}{\chi_{\pi\pi,0}}k^2 - i\omega \end{pmatrix}, \tag{20}$$

where we defined $d_\chi = \chi_{\epsilon\epsilon,0}\chi_{nn,0} - (\chi_{n\epsilon,0})^2$ the determinant of the susceptibility matrix in the $\epsilon, n$ sector. The poles of the Green's functions associated to this system are the frequencies for which $\det \hat{K} = 0$. The roots of this polynomial in the long wavelength limit are a diffusion mode (originating in charge diffusion) and two propagating sound modes

$$\omega_D = -iD_\rho^0 k^2 + \mathcal{O}(k^3) , \quad \omega_\pm = \pm c_s^0 k - \frac{i}{2}D_s^0 k^2 + \mathcal{O}(k^3) , \tag{21}$$

where the speed of sound and the two diffusion constants are defined as

$$D_\pi^0 \equiv \frac{\hat{\eta}}{\chi_{\pi\pi,0}} , \qquad (c_s^0)^2 \equiv \frac{n_0^2\chi_{\epsilon\epsilon,0} + (\chi_{\pi\pi,0})^2\chi_{nn,0} - 2n_0\chi_{\pi\pi,0}\chi_{n\epsilon,0}}{\chi_{\pi\pi,0}d_\chi} , \tag{22a}$$

$$D_\rho^0 \equiv \frac{\sigma_Q(\chi_{\pi\pi,0})^2}{n_0^2\chi_{\epsilon\epsilon,0} + (\chi_{\pi\pi,0})^2\chi_{nn,0} - 2n_0\chi_{\pi\pi,0}\chi_{n\epsilon,0}} , \qquad D_s^0 \equiv D_\pi^0 - D_\rho^0 + \sigma_Q\frac{\chi_{\epsilon\epsilon,0}}{d_\chi} . \tag{22b}$$

Note that for $n_0 = 0$ the speed of sound reduces to the familiar expression $c_s^2 = \delta P/\delta\epsilon$ (using Eq. (13) and the inverse susceptibility matrix). Similarly to what we did in the transverse sector, we can compute the retarded Green's functions by inverting the dynamical system [16]

$$\hat{G}_L^R(\omega, k) = \hat{K}^{-1} \cdot \hat{K}(\omega = 0) \cdot \chi_{L,0} = (\mathbb{1}_3 + i\omega\hat{K}^{-1}) \cdot \chi_{L,0} , \tag{23}$$

where the middle equation enforces the condition that the static $\omega = 0$ part reduces to the longitudinal part of the thermodynamic susceptibility matrix $\chi_{L,0}$. The various correlators can then be obtained

$$\hat{G}_{\epsilon\epsilon}^R(\omega, k) = \frac{k^2\chi_{\pi\pi,0}}{d(\omega, k)}\left[\frac{(c_s^0)^2\chi_{\epsilon\epsilon,0}}{\chi_{\pi\pi,0}}D_\rho^0 k^2 - i\omega\right] , \tag{24a}$$

$$\hat{G}_{nn}^R(\omega, k) = \frac{\chi_{\pi\pi,0}}{k^2 d(\omega, k)}\left[(c_s^0)^2 D_\rho^0\left(\chi_{\pi\pi,0}\chi_{nn,0}k^2 - d_\chi i\omega D_\pi^0 k^2 - d_\chi\omega^2\right) - i\omega n_0^2\right] , \tag{24b}$$

$$\hat{G}_{\pi_x\pi_x}^R(\omega, k) = \frac{k^2}{d(\omega, k)}\left[(c_s^0)^2\left(\chi_{\pi\pi,0}D_\rho^0 k^2 - i\omega(\chi_{\pi\pi,0} + D_\rho^0\chi_{\epsilon\epsilon,0}D_\pi^0 k^2)\right) - \chi_{\pi\pi,0}D_\pi^0\omega^2\right] , \tag{24c}$$

$$\hat{G}_{\epsilon n}^R(\omega, k) = \frac{k^2}{d(\omega, k)}\left[(c_s^0)^2\chi_{n\epsilon,0}D_\rho^0 k^2 - i\omega n_0\right] , \tag{24d}$$

with the normalized determinant of the dynamical matrix $d(\omega, k) = i(\omega - \omega_D)(\omega - \omega_+)(\omega - \omega_-)$. The other correlators can be obtained via the Ward identities

$$\hat{G}_{\pi_x\epsilon}^R = \frac{\omega}{k}\hat{G}_{\epsilon\epsilon}^R , \quad \hat{G}_{\pi_x n}^R = \frac{\omega}{k}\hat{G}_{\epsilon n}^R , \quad \hat{G}_{\epsilon\pi_x}^R = \frac{k}{\omega}\hat{G}_{\pi_x\pi_x}^R , \tag{25}$$

as well as the Onsager reciprocal relations

$$G_{\pi_x n}^R(\omega, k) = -G_{n\pi_x}^R(\omega, -k) , \quad G_{\pi_x\epsilon}^R(\omega, k) = -G_{\epsilon\pi_x}^R(\omega, -k) , \quad G_{\epsilon n}^R(\omega, k) = G_{n\epsilon}^R(\omega, -k) . \tag{26}$$

# 3   Hydrodynamic fluctuations in a lattice background

We shall now redo the fluctuation analysis in a lattice background. This lattice will be sourced by a periodically modulated external chemical potential

$$\mu_{\text{ext}}(x) = \mu_0 \left(1 + A \cos(Gx)\right) \ , \tag{27}$$

such that the fluid is still at rest and in local equilibrium, but all its constituents will now slowly vary in space. In particular, this last assumption of local equilibrium means that the scale of spatial fluctuations of $\mu_{\text{ext}}$ and other local quantities must be larger than the local equilibration scale. Therefore, we must have $G \lesssim \mu, T$.[8]

This lattice background manifestly breaks translation invariance. Momentum is therefore no longer a strictly conserved quantity. However, as the breaking is sourced through a hydrodynamic variable and as we assume it is weakly broken, we can still use hydrodynamic analysis [5, 6, 17, 18]. The spectral function of the associated operator to this deformation — the charge density $J^t = n$ —, evaluated in the homogeneous background, can be used to compute the momentum relaxation rate. This is known as the memory matrix formalism and was thoroughly detailed in e.g. [5, 19]. The momentum relaxation rate induced by an operator $\mathcal{O}$ sourced at wavenumber $G$ with strength $g$ takes the form [20]

$$\Gamma_{\text{mem.}}(g, G) \equiv \frac{g^2 G^2}{\bar{\chi}_{\pi\pi}} \lim_{\omega \to 0} \frac{\operatorname{Im} \hat{G}^R_{\mathcal{O}O}(\omega, k = G)}{\omega} \ . \tag{28}$$

For a cosine ionic lattice Eq. (27), $g = \mu_0 A/2$, and we have two deformation sources, one copy each at $\pm G$ — noting that the expression (28) is parity invariant in $G$. Therefore, the memory matrix relaxation rate for an ionic lattice is

$$\Gamma_{\text{ionic,mem.}} = \frac{\mu_0^2 A^2}{2} \left[ \frac{(\chi_{nn,0} - n_0 \chi_{n\epsilon,0}/\chi_{\pi\pi,0})^2}{\sigma_Q \chi_{\pi\pi,0}} + D_\pi^0 G^2 \left(\frac{\chi_{n\epsilon,0}}{\chi_{\pi\pi,0}}\right)^2 \right] \ . \tag{29}$$

It will prove useful to separate the terms according to their scaling with $G$ in this expression as $\Gamma_{\text{ionic,mem.}} = \Gamma_\eta + \Gamma_d$ with

$$\Gamma_\eta = \frac{\mu_0^2 A^2}{2} \left(\frac{\chi_{n\epsilon,0}}{\chi_{\pi\pi,0}}\right)^2 D_\pi^0 G^2 \ , \quad \Gamma_d = \frac{\mu_0^2 A^2}{2} \frac{(\chi_{nn,0} - n_0 \chi_{n\epsilon,0}/\chi_{\pi\pi,0})^2}{\sigma_Q \chi_{\pi\pi,0}} \ . \tag{30}$$

Using the Einstein relations Eq. (22a), together with $\chi_{\pi\pi,0} = \epsilon_0 + P_0$, $\chi_{\pi n,0} = n_0$ these are a convective shear drag term $\Gamma_\eta$ and an intrinsic diffusive term [6, 20, 21]

$$\Gamma_\eta = \frac{\mu_0^2 A^2}{2} \frac{\hat{\eta} G^2}{\epsilon_0 + P_0} \left(\frac{\chi_{n\epsilon,0}}{\epsilon_0 + P_0}\right)^2 \ , \quad \Gamma_d = \frac{\mu_0^2 A^2}{2} \frac{1}{\sigma_Q} \left(\frac{(\epsilon_0 + P_0)\chi_{nn,0} - n_0 \chi_{n\epsilon,0}}{\epsilon_0 + P_0}\right)^2 \ . \tag{31}$$

We will recover this same expression for the momentum relaxation time from our Bloch wave analysis. This analysis improves on the memory matrix technique by understanding how all the hydrodynamic fluctuations behave.

In a periodically modulated background, every background quantity in local thermal equilibrium $\bar{X}(x) = \bar{X}(\bar{\mu}(x), T_0)$ now admits Fourier series expansions

$$\bar{X}(x) = \sum_n e^{inGx} \bar{X}^{(n)} \ . \tag{32}$$

---

[8]For this reason our analysis does not immediately apply to graphene or other sufficiently pure semi-metals as there the scale where hydrodynamics applies is much larger than the atomic lattice scale. One would need to have a periodically undulating graphene sheet or otherwise externally imposed periodicity for this analysis to apply.

In order to apply the same method as in the previous section, we must first know how to relate perturbations of sources and responses in this new background. Because the background is static, the susceptibilities will also be static. However, because the thermodynamic quantities are position dependent and have non-vanishing Bloch modes, the susceptibilities will now also be position/momentum dependent. In principle, they depend on *two* Bloch momenta. However, in the slowly varying hydrodynamic background we may approximate them as local functions $\chi(x)$ (see also footnote 6) that follow the expansion (32).[9] The relation between perturbations in the sources and responses is then

$$\delta\phi_A(t,x) = \bar{\chi}_{AB}(x)\delta\lambda_B(t,x) \ . \tag{33}$$

The breaking of isometry by the lattice means there is no longer a decoupling between a longitudinal and transverse sector, i.e., $\phi_A, \lambda_A$ collectively denote the responses $\{\delta\epsilon, \delta n, \delta\pi_x, \delta\pi_y\}$ and the sources $\{\delta\lambda_\epsilon, \delta\lambda_n, \delta v_x, \delta v_y\}$. Both perturbations are likewise expanded on Bloch modes matching the discrete lattice symmetry

$$\delta X_A(t,x) = \sum_n \int_{-\frac{G}{2}}^{\frac{G}{2}} \mathrm{d}k \, e^{i(k+nG)x}\delta X_A^{(n)}(t,k) \ , \tag{34}$$

for $X \in \{\delta\phi_A, \delta\lambda_A\}$. As a result of the spatial dependence in the background different Bloch modes of the perturbations cross couple

$$\delta\phi_A^{(n)}(t,k) = \sum_m \bar{\chi}_{AB}^{(m)}\delta\lambda_B^{(n-m)}(t,k) \ . \tag{35}$$

The dynamical equation can then be written, after Laplace transform, as

$$\hat{K}^{(n,m)}(\omega,k) \cdot \delta\hat{\phi}^{(m)}(\omega,k) = \delta\phi^{(n)}(t=0,k) \ . \tag{36}$$

The indices $n, m$ indicate the Brillouin zones while each block $K^{(n,m)}$ is a $4 \times 4$ matrix. The diagonal blocks $\hat{K}^{(n,n)}$ correspond to the couplings between the responses in the same Brillouin zone while the off-diagonal blocks will account for coupling between different zones. These are due to the presence of the lattice and will vanish in the limit where the lattice amplitude goes to zero $A \to 0$. We will be interested in a weak lattice where the lattice amplitude $A$ is very small, and keep only terms up to order $A^2$.[10] The coupling between two modes with momenta $k + nG$ and $k + mG$ for $m > n$ will be of order $A^{m-n}$. Moreover, within perturbation theory, terms of order $A$ in the off-diagonal blocks will contribute to the same order as terms of order $A^2$ in the diagonal blocks; we can therefore drop terms of order $A^2$ and higher in the off-diagonal blocks. This also means we can consider "nearest-neighbor" interactions only – by which we mean off-diagonal terms with $m = n \pm 1$. In the long wavelength approximation we therefore can narrow our study to the three momenta $k + nG$ with $n \in \{-1, 0, 1\}$, i.e., the first three Brillouin zones. It is important to note that the diagonal terms even in the $n = 0$ Brillouin zone can still have non-trivial higher order corrections in $A$. A similar setup was already considered in [23].

---

[9] One can analyze the general behavior of two-point functions under lattice symmetries of the background [22]. Given a two-point function $G(x,y)$, one can pick a center of mass point $x = r + \delta$, $y = r - \delta$. Under the lattice symmetry, $r \to r + L$, but $\delta$ is unchanged. Then $G(x,y) = G(r = \frac{x+y}{2}, \delta) = G(r + L, \delta)$ can be expanded in Bloch modes $G(r, \delta) = \sum_n \int \mathrm{d}k G^{(n)}(k, \delta)e^{i(k+2n\pi/L)r}$. For hydrodynamic susceptibilities $\chi = G_{J^t J^t}$ we assume that they are local, i.e., we can restrict to $\delta = 0$ to leading order. In a strictly periodic background there is no structure beyond the lattice scale and hence only the $\chi^{(n)}(k = 0, \delta = 0)$ modes are non-vanishing.

[10] For a strong lattice or strong isotropy breaking the transport coefficients become tensors and this requires an independent analysis.

We will discuss this momentarily. We shall, however, first make one more simplification. It will prove more useful to use the equations in terms of the sources $\delta\lambda_A$ with

$$\hat{\mathcal{K}}^{(n,m)}(\omega, k) \cdot \delta\hat{\lambda}^{(m)}(\omega, k) = \delta\lambda^{(n)}(t = 0, k) \ , \tag{37}$$

where we can relate the two matrices using the susceptibility matrix $\chi$ by $\hat{\mathcal{K}} = \hat{K} \cdot \chi$. In this language, the $A = 0$ dynamical matrix (20) takes the form

$$\hat{\mathcal{K}} = \begin{pmatrix} -i\omega\chi_{\epsilon\epsilon,0} & -i\omega\chi_{\epsilon n,0} & ik\chi_{\pi\pi,0} & 0 \\ -i\omega\chi_{n\epsilon,0} & \sigma_Q k^2 - i\omega\chi_{nn,0} & ikn_0 & 0 \\ ik\chi_{\pi\pi,0} & ikn_0 & \hat{\eta}k^2 - i\omega\chi_{\pi\pi,0} & 0 \\ 0 & 0 & 0 & \eta k^2 - i\omega \end{pmatrix} \ . \tag{38}$$

This choice seems to a priori obfuscate the relationship between modes more than (20) due to the off-diagonal frequency dependency. However, because $\chi$ is a static matrix, the determinants of $\hat{\mathcal{K}}$ and $\hat{K}$ have the same poles in the complex frequency plane, and in the lattice case where the inverse susceptibilities present in (20) are more complicated, this form will prove clearer.

In the next few sections, we will determine this matrix $\hat{\mathcal{K}}$ in a lattice background with lattice vector $G$ for both finite $k$ momentum fluctuations and $k = 0$ momentum fluctuations to order $A^2$ in the lattice amplitude. As standard, the zeroes of its determinants will indicate the position of the dynamical modes of this system. We will then compute the conductivity as an example of how the various correlators are modified by the presence of the lattice.

## 3.1   Finite momentum aligned fluctuation spectrum

For a generic fluctuation with momentum $k$, even in the long wavelength limit, the fluctuation matrix truncated to nearest neighbor cross-coupling sufficient for the leading order in $A$ correction will be a $12 \times 12$ matrix. This is because there is no decoupling into transverse and longitudinal sectors for a generic momentum. However, if one chooses the fluctuation momentum $k$ to align with the lattice wavevector, a decoupling does occur. Choosing $k$ along a lattice vector defined to be in the $x$-direction, a parity symmetry in the $y$-direction remains. The even and odd sectors decouple into the longitudinal and transverse parts:

$$\delta Y_L = \left\{ \delta\lambda_\epsilon^{(-1)}, \delta\lambda_n^{(-1)}, \delta v_x^{(-1)}, \delta\lambda_\epsilon^{(0)}, \delta\lambda_n^{(0)}, \delta v_x^{(0)}, \delta\lambda_\epsilon^{(1)}, \delta\lambda_n^{(1)}, \delta v_x^{(1)} \right\} \ , \tag{39a}$$

$$\delta Y_T = \left\{ \delta v_y^{(-1)}, \delta v_y^{(0)}, \delta v_y^{(1)} \right\} \ . \tag{39b}$$

The dynamical matrix is then diagonal in a $9 \times 9$ and a $3 \times 3$ block.

### 3.1.1   Transverse sector

Starting with the transverse sector, the associated dynamical matrix $\hat{\mathcal{K}}_T$ is

$$\hat{\mathcal{K}}_T = \begin{pmatrix} \eta(k - G)^2 - i\omega\chi_{\pi\pi}^{(0)} & -i\omega\chi_{\pi\pi}^{(-1)} & 0 \\ -i\omega\chi_{\pi\pi}^{(1)} & \eta k^2 - i\omega\chi_{\pi\pi}^{(0)} & -i\omega\chi_{\pi\pi}^{(-1)} \\ 0 & -i\omega\chi_{\pi\pi}^{(1)} & \eta(k + G)^2 - i\omega\chi_{\pi\pi}^{(0)} \end{pmatrix} \ . \tag{40}$$

In the hydrodynamic approximation the local static susceptibility $\bar{\chi}_{\pi\pi}(x) = \frac{\partial \pi^x}{\partial v^x}(\mu(x))$ (see footnote 6 & 9) now also depends on the lattice amplitude as can be seen from its Bloch components

$$\chi_{\pi\pi}^{(0)} = \frac{G}{2\pi} \int_{-\frac{\pi}{G}}^{\frac{\pi}{G}} \mathrm{d}x\, \bar{\chi}_{\pi\pi}(\bar{\mu}(x)) \tag{41a}$$

$$= \frac{G}{2\pi} \int_{-\frac{\pi}{G}}^{\frac{\pi}{G}} \mathrm{d}x \left[ \chi_{\pi\pi,0} + \mu_0 A \cos(Gx) \frac{\partial \chi_{\pi\pi,0}}{\partial \mu_0} + \frac{\mu_0^2 A^2}{2} (\cos(Gx))^2 \frac{\partial^2 \chi_{\pi\pi,0}}{\partial \mu_0^2} + \dots \right] \tag{41b}$$

$$= \chi_{\pi\pi,0} + \frac{\mu_0^2 A^2}{4} \frac{\partial^2 \chi_{\pi\pi,0}}{\partial \mu_0^2} \tag{41c}$$

$$\equiv \chi_{\pi\pi,0} + A^2 \chi_{\pi\pi,2}^{(0)} , \tag{41d}$$

$$\chi_{\pi\pi}^{(1)} = \frac{G}{2\pi} \int_{-\frac{\pi}{G}}^{\frac{\pi}{G}} \mathrm{d}x\, e^{-iGx} \bar{\chi}_{\pi\pi}(\bar{\mu}(x)) = \frac{\mu_0 A}{2} \frac{\partial \chi_{\pi\pi,0}}{\partial \mu_0} \equiv A \chi_{\pi\pi,1}^{(1)} = A \chi_{\pi\pi,1}^{(-1)} , \tag{41e}$$

$$\chi_{\pi\pi}^{(2)} = \frac{\mu_0^2 A^2}{8} \frac{\partial^2 \chi_{\pi\pi,0}}{\partial \mu_0^2} \equiv A^2 \chi_{\pi\pi,2}^{(2)} = A^2 \chi_{\pi\pi,2}^{(-2)} = \frac{1}{2} A^2 \chi_{\pi\pi,2}^{(0)} . \tag{41f}$$

In the previous expression, we have introduced the expansion for a given Bloch mode $X^{(n)} = \sum_m X_m^{(n)} A^m$. Note that by definition, $X_0^{(0)} = X_0$ which we will keep this way.

The poles of the transverse fluctuation matrix can now be found easily, and we have

$$\omega_{T,-1} = -iD_\perp (k-G)^2 \left[ 1 - A^2 \left( \frac{\chi_{\pi\pi,2}^{(0)}}{\chi_{\pi\pi,0}} - \chi_{\pi\pi,1}^{(1)} \chi_{\pi\pi,1}^{(-1)} \frac{(k-G)^2}{G(G-2k)} \right) \right] , \tag{42a}$$

$$\omega_{T,0} = -iD_\perp k^2 \left[ 1 - A^2 \left( \frac{\chi_{\pi\pi,2}^{(0)}}{\chi_{\pi\pi,0}} - \chi_{\pi\pi,1}^{(1)} \chi_{\pi\pi,1}^{(-1)} \frac{2k^2}{(G-2k)(G+2k)} \right) \right] , \tag{42b}$$

$$\omega_{T,1} = -iD_\perp (k+G)^2 \left[ 1 - A^2 \left( \frac{\chi_{\pi\pi,2}^{(0)}}{\chi_{\pi\pi,0}} - \chi_{\pi\pi,1}^{(1)} \chi_{\pi\pi,1}^{(-1)} \frac{(k+G)^2}{G(G+2k)} \right) \right] . \tag{42c}$$

The poles remain purely diffusive, and we see that the only effect of the lattice on the transverse sector is to renormalize the shear diffusion constants $D_\perp$ at order $\mathcal{O}(A^2)$. We do see an Umklapp-like pole in the dispersion relation at the edges of the Brillouin zones $k = \pm\frac{G}{2}$. Formally, this value of $k$ is outside of the regime of validity of the expansion in small $A$. One has to resum the perturbative expansion and then one finds level repulsion, as is well known; see also the discussion at the beginning of Sec. 3.2 and footnote 11. It is distinct from conventional Umklapp, however, in that it is not level-repulsion in the dispersion (the real part of the pole in the complex frequency plane), but in the width of the fluctuation. At the edge of the Brillouin zone the width narrows and vanishes at exactly $k = \pm\frac{G}{2}$.

### 3.1.2 Longitudinal sector

The longitudinal sector is characterized by a $9 \times 9$ dynamical matrix $\hat{\mathcal{K}}_L$ of the form of $3 \times 3$ blocks

$$\hat{\mathcal{K}}_L = \begin{pmatrix} \hat{\mathcal{K}}_L^{(D)}(\omega, k-G) & \hat{\mathcal{K}}_L^{(OD)}(\omega, k-G, k) & 0 \\ \hat{\mathcal{K}}_L^{(OD)}(\omega, k, k-G) & \hat{\mathcal{K}}_L^{(D)}(\omega, k) & \hat{\mathcal{K}}_L^{(OD)}(\omega, k, k+G) \\ 0 & \hat{\mathcal{K}}_L^{(OD)}(\omega, k+G, k) & \hat{\mathcal{K}}_L^{(D)}(\omega, k+G) \end{pmatrix} . \tag{43}$$

The $\mathcal{K}_L^{(OD)}(\omega, k, p)$ block with $k < p$ belongs to the Bloch sector $n = -1$, and the one with $k > p$ to the Bloch sector $n = 1$. In a cosine lattice, however, all background quantities are parity-invariant $X^{(-n)} = X^{(n)}$, and so from here on out, we will only use the $n > 0$ expressions. The $3 \times 3$ blocks $\mathcal{K}_L^{(D)}$ and $\mathcal{K}_L^{(OD)}$ are then given by

$$
\hat{\mathcal{K}}_L^{(D)}(\omega, k) = \begin{pmatrix} \frac{\mu_0^2 A^2}{2} \sigma_Q G^2 - i\omega \chi_{\epsilon\epsilon}^{(0)} & -i\omega \chi_{n\epsilon}^{(0)} & ik\chi_{\pi\pi}^{(0)} \\ -i\omega \chi_{n\epsilon}^{(0)} & \sigma_Q k^2 - i\omega \chi_{nn}^{(0)} & ikn^{(0)} \\ ik\chi_{\pi\pi}^{(0)} & ikn^{(0)} & \hat{\eta} k^2 - i\omega \chi_{\pi\pi}^{(0)} \end{pmatrix} , \tag{44}
$$

$$
\hat{\mathcal{K}}_L^{(OD)}(\omega, k, p) = A \begin{pmatrix} -i\omega \chi_{\epsilon\epsilon,1}^{(1)} & \frac{\mu_0 \sigma_Q}{2} p(p-k) - i\omega \chi_{n\epsilon,1}^{(1)} & \frac{\mu_0}{2}(ipn_0 + ik\chi_{n\epsilon,0}) \\ \frac{\mu_0 \sigma_Q}{2} k(k-p) - i\omega \chi_{n\epsilon,1}^{(1)} & -i\omega \chi_{nn,A}^{(1)} & \frac{\mu_0}{2} ik\chi_{nn,0} \\ \frac{\mu_0}{2}(ikn_0 + ip\chi_{n\epsilon,0}) & \frac{\mu_0}{2} ip\chi_{nn,0} & -i\omega \chi_{\pi\pi,1}^{(1)} \end{pmatrix} .
$$

To leading order in $A^2$, the $n = 0$ Bloch momenta $X^{(0)}$ still have a dependency in $A$ just as in the previous section.

While difficult, it is possible to find the poles associated to this $9 \times 9$ matrix generically. For very small momentum $k = \mathcal{O}(\varepsilon^2)$ and $G = \mathcal{O}(\varepsilon)$, they take the form

$$
\omega_{D,n} = -iD_\rho^0 (nG)^2 + \frac{i}{2}\Gamma_d + \dots , \tag{45a}
$$

$$
\omega_{D,0} = -iD_\rho k^2 + \dots , \tag{45b}
$$

$$
\omega_{S,\pm,n} = \pm c_s^0 (k + nG) - \frac{i}{2} D_s^0 G^2 + \dots , \tag{45c}
$$

$$
\omega_{S,\pm,0} = -\frac{i}{2}\Gamma_{\text{ionic,mem.}} \pm c_s^0 k - \frac{i}{2} D_s^0 k^2 + \dots , \tag{45d}
$$

with $n \in \{-1, 1\}$ and "$\dots$" indicate corrections of order $\mathcal{O}(A^2 k)$ and higher. The relaxation rates $\Gamma_d, \Gamma_{\text{ionic,mem.}}$ are of order $A^2$ and equal to the memory matrix expressions given in Eqs. (31).

For large $k$ the expressions are not easy to express. However, we can use the mixing with Umklapped Bloch waves analysis to understand numerical simulations. In the longitudinal sound sector we do observe genuine level repulsion in the modified dispersion relation at the edges of the Brillouin zone. For a visualization in an explicit example later, see Fig. 8. There is thus a true sound "band gap". Sound modes with frequencies $\omega = \pm c_s G/2$ do not exist in this latticized medium. Or more precisely put, sound with wavelengths $\lambda = 2\pi k \ll G$ propagate normally with essentially unaltered speed of sound $c_s = \frac{d\omega}{dk} = c_s^0$. As the wavelength approaches the edges of the Brillouin zone, sound slows down, and right at the edge of Brillouin zone for $\lambda = (2\pi)\frac{G}{2}$, they cease to propagate as the group velocity $c_s = \frac{d\omega}{dk}|_{k=G/2} = 0$. The medium is opaque to sound at these wavelengths. Considering possible applications to condensed matter physics, we note for completeness that all these results are of course derived assuming a fixed infinitely stiff external lattice. Lattice vibrations/phonons are not taken into account. Were one to include these in the analysis, this will likely make the level repulsion and opaqueness to sound less sharp.

## 3.2 The $k = 0$ zero momentum perturbation

The $k = 0$ zero momentum is special and asks for a separate discussion. This is for three reasons. Again in the context of condensed matter physics, the $k = 0$ fluctuation describes the homogeneous responses of the system to outside probes. These are the observed macroscopic thermal and electrical conductivities, and warrant being singled

out. Secondly, we shall see that in the limit of $k \to 0$ several modes becomes degenerate. One must always be careful with accidental degeneracies. This is also the case here. The degeneracy is lifted in the presence of the lattice deformation. However, since we only consider the lattice perturbatively, this implicitly means we consider $AV_{\text{int}} \ll k$ where $V_{\text{int}}$ is a characteristic scale denoting the strength of the interactions between the Bloch modes. The degeneracy limit and the small lattice amplitude limit do not commute. We shall illustrate this in more detail below. We can still do a perturbation analysis in $A$, but this must be done from the $k = 0$ starting point separately.[11] Finally, mathematically, the $k = 0$ fluctuation is special in that at vanishing momentum, parity in the $x$-direction ($G \leftrightarrow -G$) is restored. In the 1D lattice we consider — with lattice vector in the $x$-direction — the longitudinal and transverse fluctuations at $k = 0$ therefore break up into odd and even superselection sectors under $G \leftrightarrow -G$

$$\delta Y_{L-} = \left\{ \frac{\delta\lambda_\epsilon^{(1)} - \delta\lambda_\epsilon^{(-1)}}{2i}, \frac{\delta\lambda_n^{(1)} - \delta\lambda_n^{(-1)}}{2i}, \frac{\delta v_x^{(1)} + \delta v_x^{(-1)}}{2}, \delta v_x^{(0)} \right\} , \tag{47a}$$

$$\delta Y_{L+} = \left\{ \delta\lambda_n^{(0)}, \delta\lambda_\epsilon^{(0)}, \frac{\delta\lambda_\epsilon^{(1)} + \delta\lambda_\epsilon^{(-1)}}{2}, \frac{\delta\lambda_n^{(1)} + \delta\lambda_n^{(-1)}}{2}, \frac{\delta v_x^{(1)} - \delta v_x^{(-1)}}{2i} \right\} , \tag{47b}$$

$$\delta Y_{T-} = \left\{ \frac{\delta v_y^{(1)} - \delta v_y^{(-1)}}{2i} \right\} , \tag{47c}$$

$$\delta Y_{T+} = \left\{ \delta v_y^{(0)}, \frac{\delta v_y^{(1)} + \delta v_y^{(-1)}}{2} \right\} . \tag{47d}$$

For the sake of brevity, as $k = 0$ we have suppressed all $k$ arguments in the dynamical expressions $\delta\hat{X}^{(n)}(\omega, k = 0) = \delta\hat{X}^{(n)}(\omega)$. In this basis, the overall dynamical matrix $\hat{\mathcal{K}}' = U\hat{\mathcal{K}}U^{-1}$ is diagonal by block and the dynamical equations take the form

$$\begin{pmatrix} \hat{\mathcal{K}}_{L-}(\omega) & 0 & 0 & 0 \\ 0 & \hat{\mathcal{K}}_{L+}(\omega) & 0 & 0 \\ 0 & 0 & \hat{\mathcal{K}}_{T-}(\omega) & 0 \\ 0 & 0 & 0 & \hat{\mathcal{K}}_{T+}(\omega) \end{pmatrix} \cdot \begin{pmatrix} \delta\hat{Y}_{L-}(\omega) \\ \delta\hat{Y}_{L+}(\omega) \\ \delta\hat{Y}_{T-}(\omega) \\ \delta\hat{Y}_{T+}(\omega) \end{pmatrix} = \begin{pmatrix} \delta Y_{L-}(t=0) \\ \delta Y_{L+}(t=0) \\ \delta Y_{T-}(t=0) \\ \delta Y_{T+}(t=0) \end{pmatrix} , \tag{48}$$

where $U = \begin{pmatrix} U_L & 0 \\ 0 & U_T \end{pmatrix}$ is the matrix that reorders the fields from the basis in Eq. (39) to Eq. (47)

$$U_L = \left( \begin{array}{ccc|ccc|ccc} \frac{i}{2} & 0 & 0 & 0 & 0 & 0 & -\frac{i}{2} & 0 & 0 \\ 0 & \frac{i}{2} & 0 & 0 & 0 & 0 & 0 & -\frac{i}{2} & 0 \\ 0 & 0 & \frac{1}{2} & 0 & 0 & 0 & 0 & 0 & \frac{1}{2} \\ \hline 0 & 0 & 0 & 0 & 0 & 1 & 0 & 0 & 0 \\ 0 & 0 & 0 & 0 & 1 & 0 & 0 & 0 & 0 \\ 0 & 0 & 0 & 1 & 0 & 0 & 0 & 0 & 0 \\ \hline \frac{1}{2} & 0 & 0 & 0 & 0 & 0 & \frac{1}{2} & 0 & 0 \\ 0 & \frac{1}{2} & 0 & 0 & 0 & 0 & 0 & \frac{1}{2} & 0 \\ 0 & 0 & \frac{i}{2} & 0 & 0 & 0 & 0 & 0 & -\frac{i}{2} \end{array} \right) , \quad U_T = \begin{pmatrix} \frac{i}{2} & 0 & -\frac{i}{2} \\ 0 & 1 & 0 \\ \frac{1}{2} & 0 & \frac{1}{2} \end{pmatrix} . \tag{49}$$

---

[11] A simple example that illustrates the point is the toy model fluctuation matrix

$$\hat{K}_{\text{toy}} = \begin{pmatrix} E - k & AV_{\text{int}} \\ AV_{\text{int}} & E + k \end{pmatrix} \tag{46}$$

This has poles at $E = \pm\sqrt{k^2 + A^2 V_{\text{int}}^2}$ signaling level repulsion at $k = 0$. Expanding these poles in $A$ gives $E = \pm k(1 + \frac{1}{2}\frac{A^2 V_{\text{int}}^2}{k^2})$, whereas expanding in $k$ gives $E = \pm AV_{\text{int}}(1 + \frac{1}{2}\frac{k^2}{A^2 V_{\text{int}}^2})$.

### 3.2.1 Transverse sector

Let us again consider the transverse sector first. The dynamical matrices $\hat{\mathcal{K}}_{T-}$ and $\hat{\mathcal{K}}_{T+}$ are

$$
\hat{\mathcal{K}}_{T-} = \left( \eta G^2 - i\omega \chi_{\pi\pi}^{(0)} \right) \ , \quad \hat{\mathcal{K}}_{T+} = \begin{pmatrix} -i\omega\chi_{\pi\pi}^{(0)} & -2i\omega A\chi_{\pi\pi,1}^{(1)} \\ -i\omega A\chi_{\pi\pi,1}^{(1)} & \eta G^2 - i\omega\chi_{\pi\pi}^{(0)} \end{pmatrix} \ , \tag{50}
$$

These have the following diffusive poles

$$
\omega^{(T-)} = -iD_\perp G^2 \left[ 1 - A^2 \frac{\chi_{\pi\pi,2}^{(0)}}{\chi_{\pi\pi,0}} \right] \ , \tag{51a}
$$

$$
\omega_0^{(T+)} = 0 \ , \tag{51b}
$$

$$
\omega_1^{(T+)} = -iD_\perp G^2 \left[ 1 - A^2 \frac{\chi_{\pi\pi,2}^{(0)}}{\chi_{\pi\pi,0}} + 2A^2 \left( \frac{\chi_{\pi\pi,1}^{(1)}}{\chi_{\pi\pi,0}} \right)^2 \right] \ . \tag{51c}
$$

There are several aspects to note: firstly, as mentioned above these poles do not correspond to the $k \to 0$ limit of the finite $k$ fluctuations in Eq. (42). The indicated emergent degeneracy at $k \to 0$ between $\omega_{T,-1}$ and $\omega_{T,1}$ is obvious in Eq. (42). The lattice perturbation $A$ lifts this degeneracy and therefore the limits $k \to 0$ and $A \to 0$ do not commute. Secondly, there is a zero mode in the $T+$-sector. This is the standard transverse $k = 0$ excitation, that corresponds to a change of the static homogeneous transverse pressure background and as a zero mode should not be considered in the fluctuation spectrum.

### 3.2.2 Longitudinal sector

Consider the $G$-parity odd longitudinal sector first. Its dynamical matrix $\hat{\mathcal{K}}_{L-}$ is given by

$$
\hat{\mathcal{K}}_{L-} = \left( \begin{array}{ccc|c} & & & \frac{\mu_0 AG}{2}\chi_{\epsilon n,0} \\ & \mathcal{L}_- & & \frac{\mu_0 AG}{2}\chi_{nn,0} \\ & & & -i\omega A\chi_{\pi\pi,1}^{(1)} \\ \hline -\mu_0 A\chi_{n\epsilon,0}G & \mu_0 A\chi_{nn,0}G & -2i\omega A\chi_{\pi\pi,1}^{(1)} & -i\omega\chi_{\pi\pi}^{(0)} \end{array} \right) \ , \tag{52}
$$

with

$$
\mathcal{L}_- = \begin{pmatrix} -i\omega\chi_{\epsilon\epsilon}^{(0)} + \frac{\mu_0^2 A^2}{2}\sigma_Q G^2 & -i\omega\chi_{\epsilon n}^{(0)} & \chi_{\pi\pi}^{(0)}G \\ -i\omega\chi_{n\epsilon}^{(0)} & \sigma_Q G^2 - i\omega\chi_{nn}^{(0)} & n^{(0)}G \\ -\chi_{\pi\pi}^{(0)}G & -n^{(0)}G & \hat{\eta}G^2 - i\omega\chi_{\pi\pi}^{(0)} \end{pmatrix} \ . \tag{53}
$$

The lines are there to highlight the coupling between two sub-sectors. The top-left block is equivalent to the coupling matrix (20) in the homogeneous system at momentum $G$ encoding the $G$-parity odd part of two sound modes and a charge diffusion mode (see Eq. (21)), while the lower-right block reflects the conservation of momentum in the homogeneous case. In the presence of the lattice deformation the momentum conservation mode now couples with the Umklapped finite $G$ sound-, and charge diffusion modes through the off-diagonal terms. We can find the modes of this dynamical matrix in the same way we

did before, and we find

$$\omega_{\text{Drude}}^{(L-)} = -i(\Gamma_d + \Gamma_\eta) \, , \tag{54a}$$

$$\omega_D^{(L-)} = -i\left(D_\rho^0 + A^2 D_\rho^{(L-),2}\right)G^2 + i\Gamma_d \, , \tag{54b}$$

$$\omega_\pm^{(L-)} = \pm\left(c_s^0 + A^2 c_s^{(L-),2}\right)G - \frac{i}{2}\left(D_s^0 + A^2 D_s^{(L-),2}\right)G^2 \, . \tag{54c}$$

We therefore see that the poles in (54) are those of the homogeneous system (21) at momentum $G$ with corrected diffusion constants due to the effects of the lattice. The procedure to compute the explicit form of the corrections $D_\rho^{(L-),2}$, $c_s^{(L-),2}$, $D_s^{(L-),2}$ is detailed in Appendix C. For a generic relativistic fluid these are quite involved; in the explicit case of a fluid with conformal invariance they simplify greatly and we give the expressions below in Eq. (61).

The most noteworthy part is the Drude pole $\omega_{\text{Drude}}$. As previewed at the beginning of this Sec. 3 the lattice breaks translational symmetry and this induces a momentum decay rate. The more detailed Bloch wave analysis which gives us all fluctuations at finite $k, \omega$ in the hydrodynamic regime beautifully recovers the finite $\omega$ memory matrix result (29), as it should. In the condensed matter context, it is this sector specifically that governs the $k = 0$ thermoelectric conductivities, where the presence of the second diffusive mode $\omega_D$ (originating in Umklapped charge diffusion) in addition to the Drude mode has significant observable consequences as expounded in [10].

For the $G$-parity even sector the dynamical matrix is given by

$$\hat{\mathcal{K}}_{L+} = \left(\begin{array}{cc|ccc} -i\omega\chi_{nn}^{(0)} & -i\omega\chi_{n\epsilon}^{(0)} & -2i\omega A\chi_{n\epsilon,1}^{(1)} & -2i\omega A\chi_{nn,1}^{(1)} & 0 \\ -i\omega\chi_{n\epsilon}^{(0)} & -i\omega\chi_{\epsilon\epsilon}^{(0)} + \frac{\mu_0^2 A^2}{2}\sigma_Q G^2 & -2i\omega A\chi_{\epsilon\epsilon,1}^{(1)} & A(\mu_0\sigma_Q G^2 - 2i\omega\chi_{n\epsilon,1}^{(1)}) & -\mu_0 AGn_0 \\ \hline -i\omega A\chi_{n\epsilon,1}^{(1)} & -i\omega A\chi_{\epsilon\epsilon,1}^{(1)} & & & \\ -i\omega A\chi_{nn,1}^{(1)} & A(\frac{\mu_0\sigma_Q}{2}G^2 - i\omega\chi_{n\epsilon,1}^{(1)}) & & \mathcal{L}_+ & \\ 0 & \frac{\mu_0 A}{2}Gn_0 & & & \end{array}\right) \, , \tag{55}$$

with

$$\mathcal{L}_+ = \begin{pmatrix} \frac{\mu_0^2 A^2}{2}\sigma_Q G^2 - i\omega\chi_{\epsilon\epsilon}^{(0)} & -i\omega\chi_{n\epsilon}^{(0)} & -\chi_{\pi\pi}^{(0)}G \\ -i\omega\chi_{n\epsilon}^{(0)} & \sigma_Q G^2 - i\omega\chi_{nn}^{(0)} & -n^{(0)}G \\ \chi_{\pi\pi}^{(0)}G & n^{(0)}G & \hat{\eta}G^2 - i\omega\chi_{\pi\pi}^{(0)} \end{pmatrix} \tag{56}$$

Once again, the lines show how the various sub-sectors couple through the off-diagonal $A$-dependent terms. The $\mathcal{L}_+$ sector is again a pair of sound modes and a charge diffusion mode, but now the part that is even under $G$-parity. The $n = 0$ charge and energy conservation modes are reflected in the upper-left blocks. The modes of this system are

$$\omega_c^{(L+)} = 0 \, , \tag{57a}$$

$$\omega_d^{(L+)} = 0 \, , \tag{57b}$$

$$\omega_D^{(L+)} = -i\left(D_\rho^0 + A^2 D_\rho^{(L+),2}\right)G^2 \, , \tag{57c}$$

$$\omega_\pm^{(L+)} = \pm\left(c_s^0 + A^2 c_s^{(L+),2}\right)G - \frac{i}{2}\left(D_s^0 + A^2 D_s^{(L+),2}\right)G^2 \, , \tag{57d}$$

We see again that the latter three poles in (57) are those of the homogeneous system (21) at momentum $G$ with corrected diffusion constants due to the effects of the lattice. The shifts differ, however, from the $L-$ sector. The explicit form of the corrections $D_\rho^{(L+),2}$, $c_s^{(L+),2}$, $D_s^{(L+),2}$ can again be derived through the method detailed in Appendix C, and tractable expressions for the special case of a conformal fluid are given in Eq. (61). The first two poles are the ones encoding charge conservation and energy conservation; they remain unshifted at this order in perturbation theory. As we explain in Appendix C, one of these modes is an exact conservation mode and by rotating the system back to $\hat{K}$ instead of $\hat{\mathcal{K}}$, it is easy to see that this mode corresponds to overall charge conservation — associated here to $\delta\lambda_n^{(0)}$. The other mode is merely unshifted at this order in perturbation theory.

Having computed the corrections to the $k = 0$ modes, one clearly sees that these are not equal to the limit $k \to 0$ of (45). In that limit, at leading order, $\omega_{D,\pm 1} = -iD_\rho^0 G^2 + \frac{i}{2}\Gamma_d$ and $\omega_{S,\pm,0} = -\frac{i}{2}\Gamma_{\text{ionic,mem.}}$, whereas the explicit $k = 0$ computation has $\omega_{\text{Drude}}^{(L-)} = -\Gamma_{\text{ionic,mem.}}$, and an additional (to leading order in $A^2$) zero mode. This difference is due to the non-commutativity of the $k \to 0$ and $A \to 0$ limits. In the next section, where we illustrate the emergence of these hydrodynamic modes in an explicitly computed example, we will show precisely how these poles are related in the $k \to 0$ limit.

# 4 Bloch wave hydrodynamics emerging from holographic models: a comparison

We will now validate the understanding of charged (relativistic) hydrodynamics in a periodic potential by comparing it with the low energy physics of holographic models. Holographic models describe the strong coupling regime of quantum field theories in a manifestly real time formalism. Uniquely so, this includes the emergence of hydrodynamics at low frequencies and long wavelengths $\omega, k \ll T, \mu$ [24–27]. This last part is also known as fluid-gravity duality [28]. By considering a strongly coupled quantum field theory in a spatially periodic chemical potential background, i.e., an ionic lattice[12], described holographically in terms of its dual gravitational description, we will see that the Bloch wave hydrodynamics described above emerges. There is one simplifying feature in the two holographic models we choose here. Both describe a conformally invariant system for which the equation of state takes the scaling form

$$P(T, \mu) = T^{d+1} f(T/\mu) \, , \tag{58}$$

which directly implies $P = \epsilon/d$. Furthermore, due to the conformal symmetry such a system must also have a vanishing bulk viscosity, i.e., $\zeta = 0$.

The two specific models we consider are the strongly coupled theories holographically dual to the Reissner-Nordström (RN) black hole [29–33] as well as the Gubser-Rocha (GR) black hole [34]. We will solve the fluctuations in these systems modulated by a finite chemical potential numerically and compare to the predictions from Bloch wave hydrodynamics as presented in the previous sections. These two systems are chosen as their ground states are possible candidates to explain the mysterious strange metal physics underlying high $T_c$ superconductors. There is reason to believe that this physics is indeed that of strongly coupled electrons in an ionic lattice. The possible relevance of Bloch

---

[12]This mimics the charge distribution of a frozen atomic lattice, or more appropriately an ionic lattice with valence electrons.

wave hydrodynamics in the context of strange metal physics is described in a companion article [10].

A brief description of the numerical holographic set-up is provided in appendix D.2; more details can be found in [10]. The crucial aspect of relevance here to the comparison of the numerics with our Bloch hydrodynamic analysis is the equation of state of the two models. The $2 + 1$ dimensional (finite temperature) field theories dual to AdS$_4$ RN and GR black holes are conformal charged fluids[13] with equation of state

$$\frac{P_{\rm RN}(\hat{T})}{\mu^3} = \hat{T}^3 \left( \frac{-1 - 8\pi^2 \hat{T}^2 + 2\pi \hat{T}\sqrt{3 + 16\pi^2 \hat{T}^2}}{2\hat{T}^3(4\pi\hat{T} - \sqrt{3 + 16\pi^2 \hat{T}^2})^3} \right) \ , \tag{59}$$

$$\frac{P_{\rm GR}(\hat{T})}{\mu^3} = \hat{T}^3 \left( \frac{(3 + 16\pi^2 \hat{T}^2)^{3/2}}{27\hat{T}^3} \right) \ , \tag{60}$$

where we have defined $\hat{T} \equiv T/\mu$.

A direct consequence of this conformal equation of state is that $\chi_{\epsilon\epsilon,0} = d\,\chi_{\pi\pi,0}$ and $\chi_{n\epsilon,0} = d\,n_0$. As a consequence many previous expressions simplify. Specifically the order $A^2$ corrections to the poles for $k = 0$ are now given by the tractable expressions:

$$D_\rho^{(L-),2} = -\frac{\sigma_Q}{\chi_{\pi\pi,0}} \frac{\chi_{nn,2}^{(0)}(\chi_{\pi\pi,0})^3 - 2\chi_{nn,1}^{(1)}(\chi_{\pi\pi,0})^2\mu_0 n_0 + \chi_{nn,0}\chi_{\pi\pi,0}\mu_0^2 n_0^2 + \mu_0^2 n_0^4}{(d_\chi)^2} \ , \tag{61a}$$

$$D_\rho^{(L+),2} = \frac{\sigma_Q}{4\chi_{\pi\pi,0}} \frac{(\chi_{\pi\pi,0})^2\left((\chi_{nn,0})^2\mu_0^2 - 4\chi_{nn,2}^{(0)}\chi_{\pi\pi,0}\right) + 8\chi_{nn,1}^{(1)}(\chi_{\pi\pi,0})^2\mu_0 n_0 - 12\chi_{nn,0}\chi_{\pi\pi,0}\mu_0^2 n_0^2 + 12\mu_0^2 n_0^4}{(d_\chi)^2} \ , \tag{61b}$$

$$c_s^{(L-),2} = \frac{\mu_0^2 n_0^2}{4\sqrt{2}(\chi_{\pi\pi,0})^2} \ , \qquad D_s^{(L-),2} = \frac{\mu_0^2}{4\chi_{\pi\pi,0}} \left[ \sigma_Q + \frac{10n_0^2 - 3\chi_{nn,0}\chi_{\pi\pi,0}}{(\chi_{\pi\pi,0})^2}\hat{\eta} \right] \ , \tag{61c}$$

$$c_s^{(L+),2} = \frac{\mu_0^2 n_0^2}{\sqrt{2}(\chi_{\pi\pi,0})^2} \ , \qquad D_s^{(L+),2} = \frac{\mu_0^2}{4\chi_{\pi\pi,0}} \left[ \sigma_Q - \frac{3\chi_{nn,0}}{(\chi_{\pi\pi,0})^2}\hat{\eta} \right] \ . \tag{61d}$$

The explicit expressions for the thermodynamic quantities in the grand canonical ensemble for the RN and GR black holes can be found in Appendix D.1.

We will use the longitudinal optical conductivity $\sigma_{xx}(\omega, k_x = k) = \frac{1}{i\omega}\langle J^x(-\omega, -k_x)J^x(\omega, k_x)\rangle$ as a probe. Generically this current will receive contributions from all hydrodynamic fluctuations; these essentially determine the low frequency long-wavelength response. At finite $k$, this means we should see all 9 modes described in (45). At $k = 0$, however, the current is part of the $L-$ sector and we will only see the first sector with its 4 modes. Fig. 1 gives a schematic sketch of what the spectrum of the current-current correlator — and therefore the optical conductivity — should look like in the complex frequency plane based on our hydrodynamic predictions.

Precisely this expectation is reproduced by the numerical results in holographic duals to RN and GR black holes where hydrodynamics is emergent. Fig. 2 plots the density of the absolute value and argument of the optical conductivity for small values of the real part of the frequencies, i.e., zoomed in near the imaginary frequency axis. Each picture is at a different value of $k/\mu \in \{0, 0.001, 0.005\}$. We see that for $k = 0$ there are only two purely diffusive poles, as predicted, that split into two propagating and two diffusive poles at finite $k$. For finite $k$, there should also be a third diffusive pole very close to the real

---

[13]While this result is well-known for the RN black hole, it only applies in the GR black hole for a suitable choice of quantization of the boundary scalar operator — the dilaton must be a marginal deformation. One must therefore use mixed boundary conditions for the dilaton at the boundary [35].

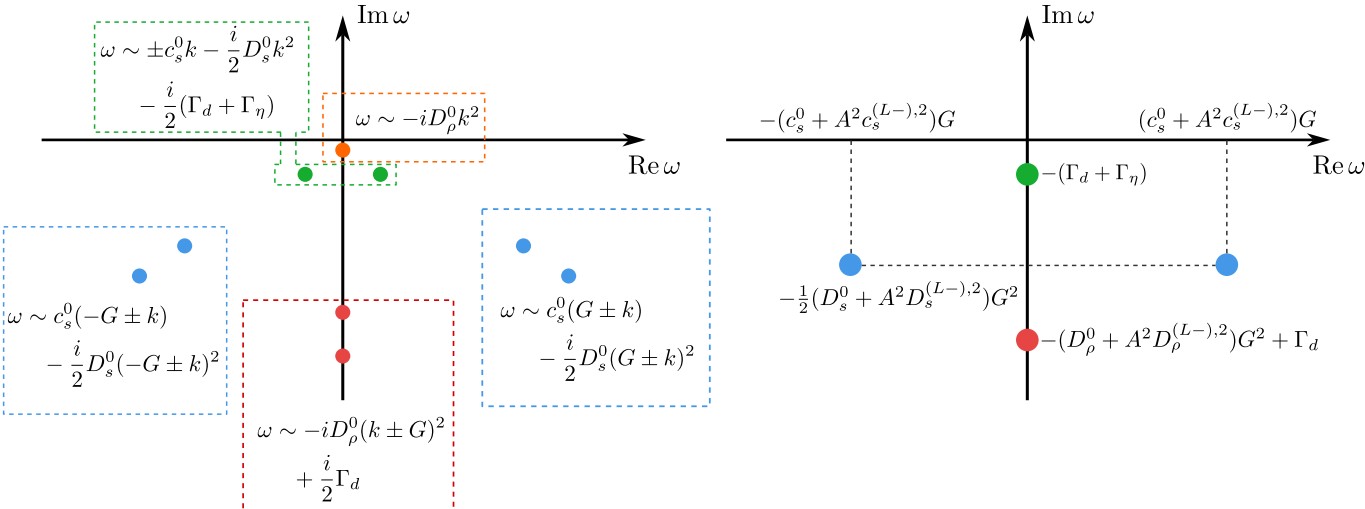

Figure 1: Drawing of the expected position of the poles in the current-current correlator at finite $k$ (left) and at $k = 0$ (right) based on the hydrodynamical predictions in terms of Bloch waves in Sec. 3. Generically there are 9 poles: the standard two sound modes plus a (charge) diffusive mode of charged hydrodynamics cross coupled with the $n = 1$ and $n = -1$ Umklapp copies of each. At $k = 0$, there is an emergent symmetry due to which the longitudinal current-current correlator only probes the first sector $\hat{\mathcal{K}}_{L-}$ that contains the 4 modes that are odd under inverting the lattice momentum $G \leftrightarrow -G$.

axis. Its weight is very low, however, but it can be unveiled by zooming in carefully. This was plotted in Fig. 3 for $k/\mu \in \{0.006, 0.008, 0.01\}$ (this choice of momenta proved more convenient to display). In all cases the location of these poles can be compared with the predictions from our hydrodynamical analysis after substituting in the relevant equation of state. The match is perfect for both $k = 0$ and $k$ finite as denoted by the white circles in Fig. 2 and the triangles in Fig. 3. Similarly, we plotted in Fig. 4 the argument of the optical conductivity near the sound poles at momenta $G \pm k$. The weight of these poles is very small and they are therefore difficult to identify in $|\sigma|$. The argument of $\sigma$, on the other hand, displays a jump at the poles. A similar analysis holds for the conjugate pair of poles at $-G \pm k$. These sound poles give rise to a characteristic peak in the real conductivity, first noted in [36–38].

To illustrate in more detail the hydrodynamical origin of all these poles and their full explanation in terms of thermodynamic quantities, one can track the location of the poles as a function of temperature. Focusing only on the purely diffusive poles (two for $k = 0$, and three for $k \neq 0$), as they are more easily extracted numerically by scanning carefully over the negative imaginary frequency axis, we also find here a perfect match between numerics and hydrodynamic prediction, Eqs. (54) and (45) respectively, but now as function of $T/\mu$; see Fig. 5 and Fig. 6.

With our computational RN and GR examples we can also illustrate the subtle nature of the $k \to 0$ limit. As we saw in the previous section, the naive extrapolation to $k \to 0$ of the two $n = 0$ sound modes $\omega_{S,0,\pm} = -\frac{i}{2}\Gamma_{\text{ionic,mem.}} + \mathcal{O}(k)$ does not correspond with the physical $k = 0$ Drude pole $\omega_{\text{Drude}}^{(L-)} = -i\Gamma_{\text{ionic,mem.}}$ and its $(L+)$ equivalent $\omega_d^{(L+)} = 0$. To emphasize this once more, the origin of this difference comes from the non-commutativity of the $k \to 0$ and $A \to 0$ limits. In Fig. 7, we have carefully analyzed the low $k$ regime of the GR black hole. For $k/\mu = 10^{-4}$, the two diffusive poles are close to their $k = 0$ values (54) and (57). As we increase $k$, they get closer and collide, leading to the two

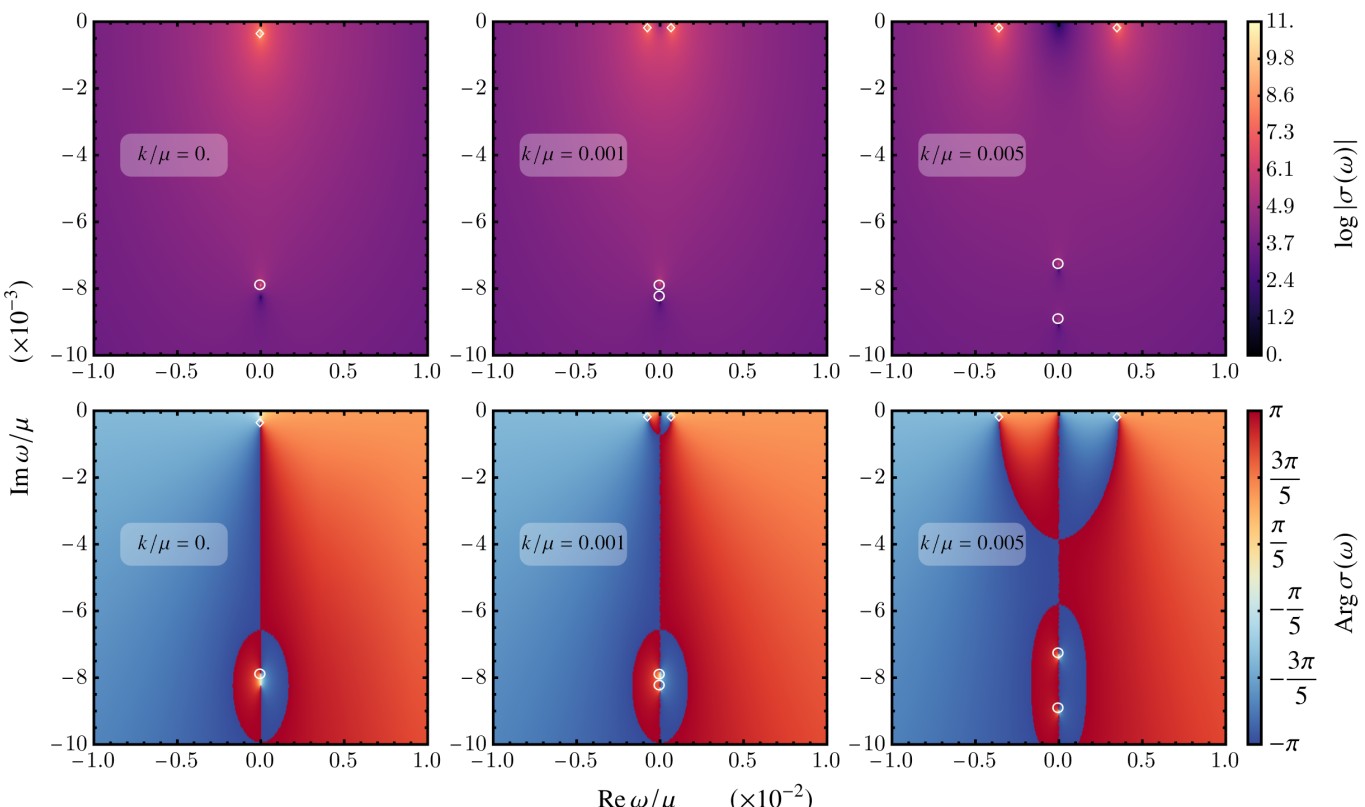

Figure 2: Density plot of (the logarithm of) the GR longitudinal conductivity $\log |\sigma(\omega)|$ (top) and its argument $\mathrm{Arg}\,\sigma(\omega)$ (bottom) in the complex frequency plane close to the imaginary axis for $T/\mu = 0.1$, $A = 0.05$ and $G/\mu = 0.1$, for four values $k/\mu \in \{0, 0.001, 0.005, 0.01\}$. At $k = 0$, we see the poles $\omega_{\mathrm{Drude}}^{(L-)}$ and $\omega_D^{(L-)}$ while at $k > 0$, the Drude pole splits into the two sound modes $\omega_{S,\pm,0}$ (denoted by a white $\diamond$) and the diffusion pole splits into $\omega_{D,\pm 1}$ (denoted by a white $\circ$). The markers indicate the analytical position of these poles prescribed by our hydrodynamical derivation. A priori, a fifth pole $\omega_{D,0}$ at $k > 0$ also couples to the electrical current, but it is not visible on the range plotted. A more refined computation, does reveal it (Fig. 3).

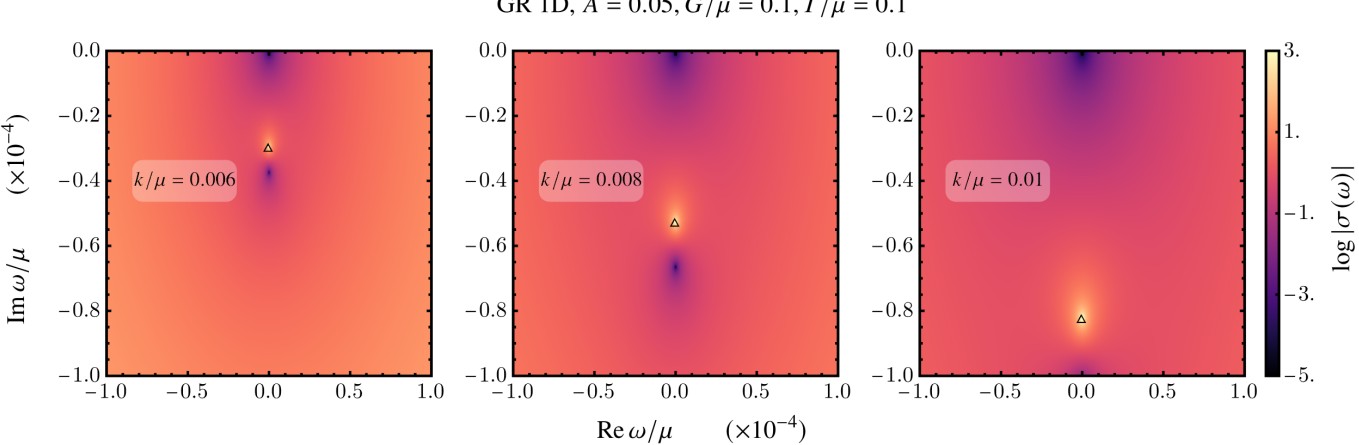

Figure 3: GR conductivity $\log |\sigma(\omega)|$ in the complex plane close to the imaginary axis for $T/\mu = 0.1$, $A = 0.05$ and $G/\mu = 0.1$, varying $k/\mu \in \{0.006, 0.008, 0.01\}$. We see a purely diffusive pole on the imaginary axis which matches the hydrodynamic diffusion pole $\omega_{D,0}$ (denoted by $\triangle$). The area plotted is zoomed on the origin compared to Fig. 2. There the diffusive pole was too small to be visible.

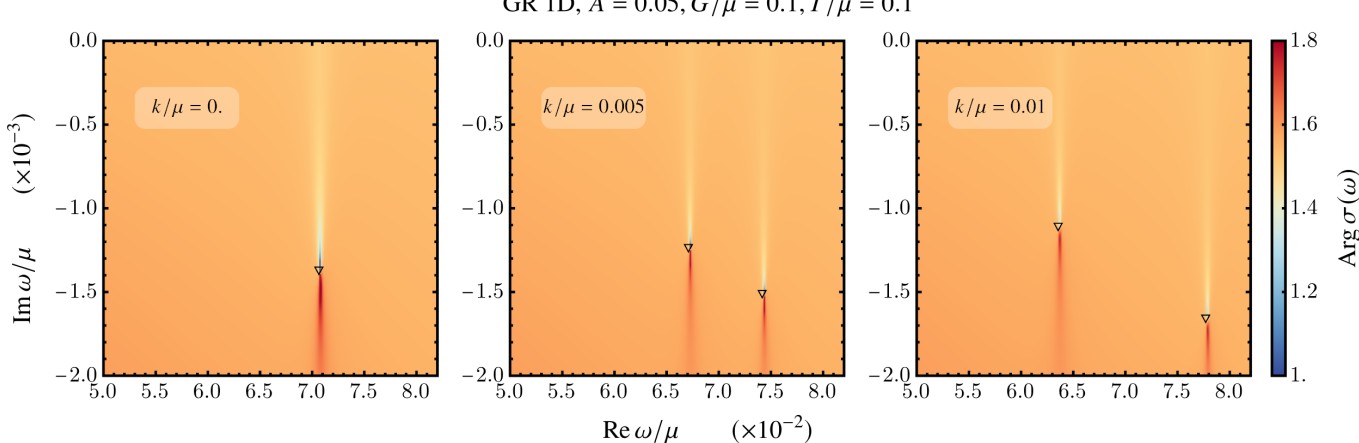

Figure 4: Argument of the GR conductivity arg $\sigma(\omega)$ in the complex plane for $T/\mu = 0.1$, $A = 0.05$ and $G/\mu = 0.1$, varying $k/\mu \in \{0, 0.001, 0.005, 0.01\}$. At $k = 0$, we see the sound pole $\omega_+^{(L-)}$. Its real part is precisely at $c_s G$ with $c_s = 1/\sqrt{2}$ in a $d = 2$ conformal fluid. At $k > 0$, it splits into the two sound modes $\omega_{S,\pm,1}$ (denoted by $\triangledown$). The markers indicate the analytical position of these poles prescribed by our hydrodynamical derivation. These poles are more difficult to observe in $|\sigma|$ than those on the imaginary axis and are easier to see as jumps in the argument of the complex function.

sound modes of (45). This diffusion-to-sound crossover happens when $\frac{k}{G} \sim A^2$ illustrating the non-commuting limits $k \to 0$, $A \to 0$ which means we can estimate the characteristic length scale of the interactions to be $V_{\text{int}} \sim AG$ (see footnote 11).

Finally, to re-emphasize the underlying Bloch wave Umklapp physics, Fig. 8 shows the real part of the momentum-dependent optical conductivity $\sigma(\omega, k)$ in the $\omega, k$ plane. The right-hand plot is a zoomed-in version of the left-hand plot near the edge of the Brillouin zone $k = \frac{G}{2}$. The gray dots are numerically obtained solutions of $\det \hat{\mathcal{K}}_L = 0$ for the same parameters, showing that the hydrodynamic description of the matrix (43) at order $\mathcal{O}(A^2)$ matches the data over the entire Brillouin zone. At low frequency, we see the expected sound mode $\omega \sim c_s k$ dominating the low frequency regime, but we also can see its interaction with the sound mode $\omega \sim c_s(G-k)$. They meet at the edge of the Brillouin zone $k = \frac{G}{2}$ and in the right-hand plot, we see the traditional level repulsion of Umklapp and the opening of a gap in the sound mode spectrum.

## 5 Conclusion

The crucial message of this paper is that hydrodynamic fluctuations in a periodically modulated background should be understood based on a Bloch wave analysis instead of simple plane waves. If the typical length scale of this modulation is sufficiently large and the amplitude sufficiently small, we can still use hydrodynamics to study the long-time response of the conserved charges. This is an old observation in neutral hydrodynamics, but deserves restudy for charged hydrodynamics given the novel experimental progress of observed hydrodynamic flow in electronic condensed matter systems [7–9]. This is particularly so in the presence of a charged fluid, which introduces an additional intrinsic diffusive mode. The presence of a spatial periodic modulation introduces Brillouin zone copies, also for this additional mode, and due to Umklapp at the Brillouin zone boundary

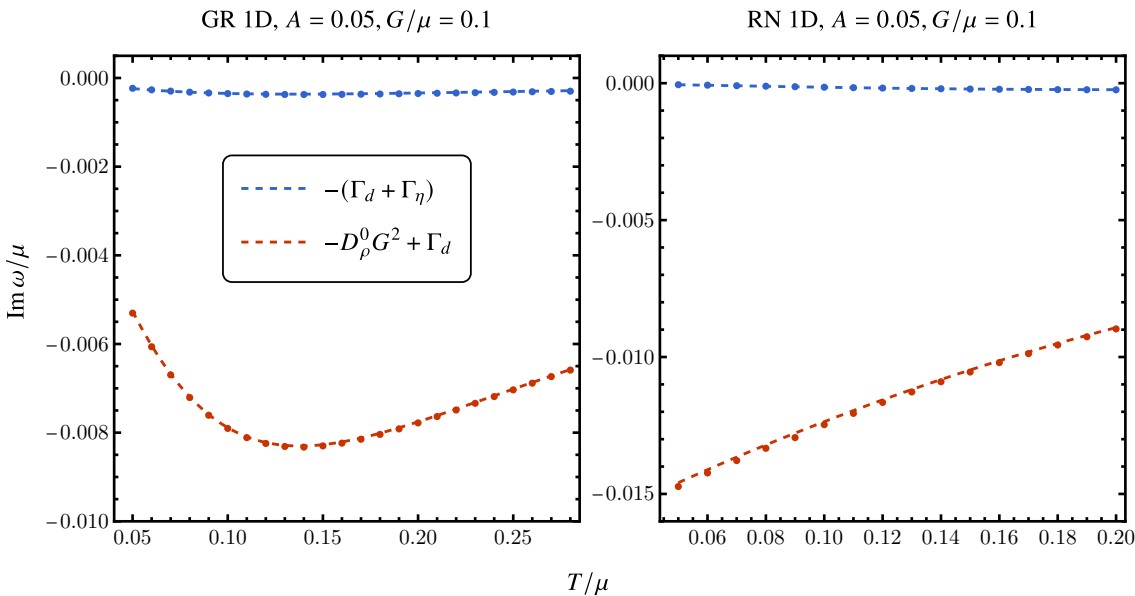

Figure 5: Comparison between the position of the poles on the imaginary axis (points) and the analytical hydrodynamical formula (54) at $k = 0$, as a function of $T/\mu$, and for $A = 0.05$ and $G/\mu = 0.1$. This is done for GR on the left and RN on the right. The blue data is the Drude pole $\omega_{\text{Drude}}^{(L-)}$ and the red data corresponds to the Umklapped diffusion pole $\omega_D^{(L-)}$ (Eqs. (54)). The corrections to the diffusion constants are smaller than our numerical accuracy for our choice of parameters, so we can simply ignore them here.

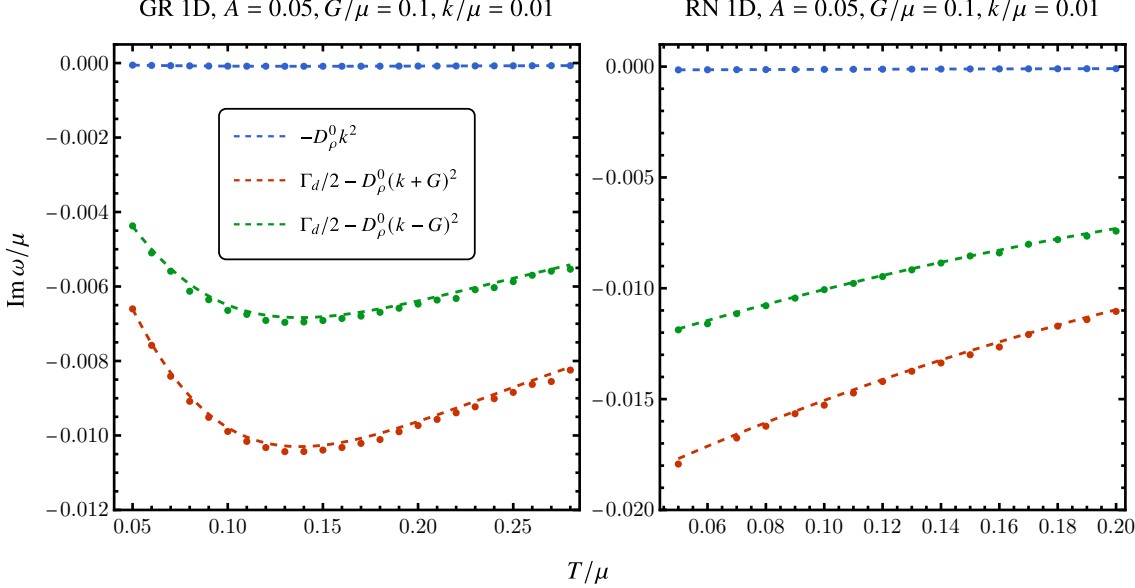

Figure 6: Comparison between the position of the poles $\omega_{D,0}$ and $\omega_{D,\pm 1}$ on the imaginary axis (points) and the analytical hydrodynamical expressions (45) at $k/\mu = 0.01$, as a function of $T/\mu$, and for $A = 0.05$ and $G/\mu = 0.1$. This is done for GR on the left and RN on the right. The corrections to the diffusion constants are smaller than our numerical accuracy for our choice of parameters, so we can simply ignore them here.

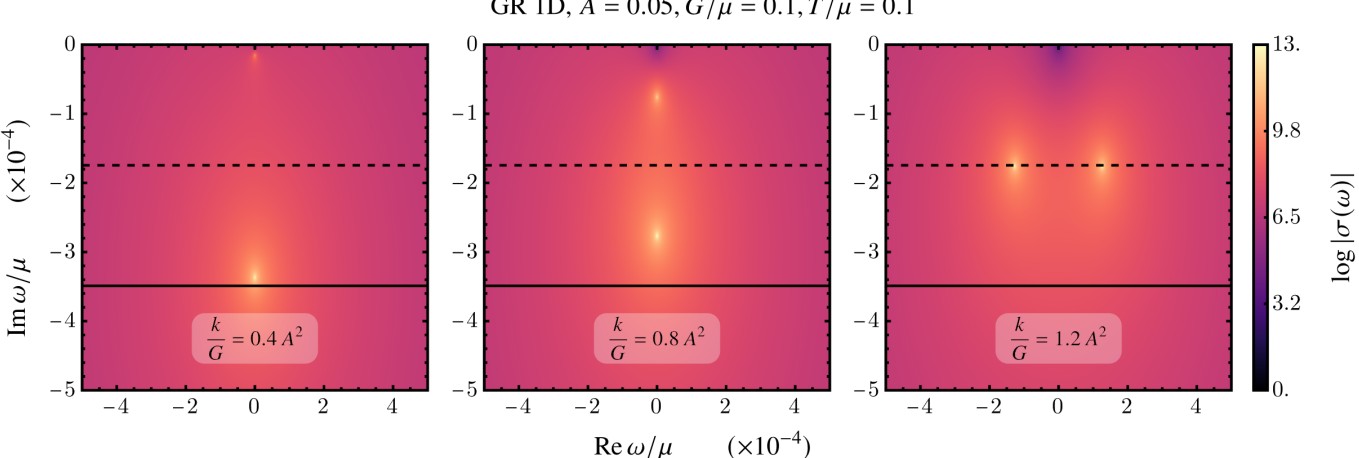

Figure 7: (Logarithm of the) GR conductivity $\log|\sigma(\omega)|$ in the complex plane close to the imaginary axis for $T/\mu = 0.1$, $A = 0.05$ and $G/\mu = 0.1$, varying $k/\mu \in \{10^{-4}, 2 \times 10^{-4}, 3 \times 10^{-4}, 4 \times 10^{-4}\}$. For $k = 10^{-4}$, we see the poles $\omega_{\text{Drude}}^{(L-)}$ and $\omega_{\text{d}}^{(L+)}$ with small corrections. For $k > 3 \times 10^{-4}$, the poles are now the two sound modes $\omega_{S,\pm,0}$ close to their $k \to 0$ limit. The lines indicate the positions $\omega_{\text{Drude}}^{(L-)} = -i\Gamma_{\text{ionic,mem.}}$ (solid) and $\omega_{S,0,\pm}(k \to 0) = -\frac{i}{2}\Gamma_{\text{ionic,mem.}}$ (dashed). When expressed in terms of $\frac{k}{G}$, the transition appears to happen at $\frac{k}{G} \sim A^2$.

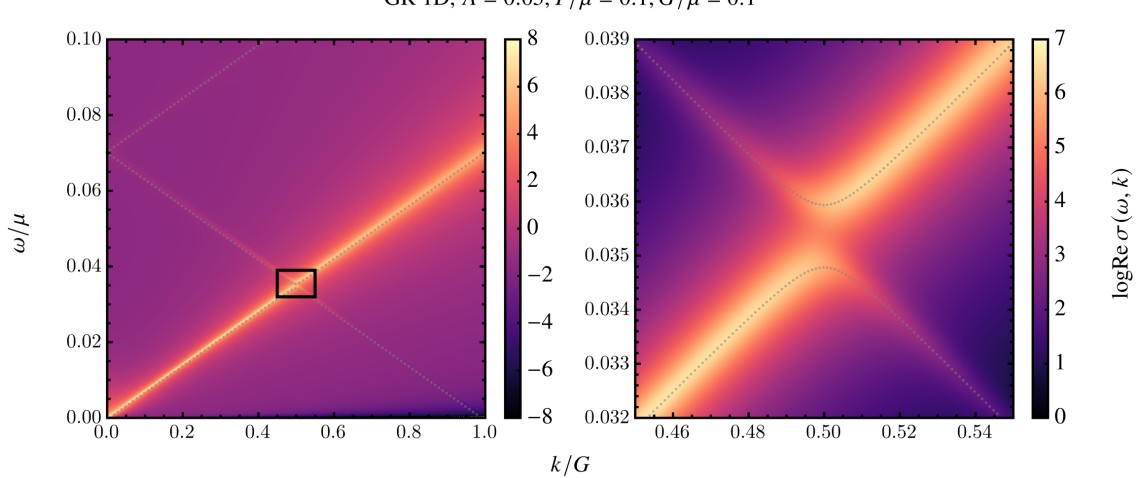

Figure 8: (Left) Re $\sigma(\omega, k)$ plotted in the $(k/G, \omega/\mu)$ plane for $A = 0.05$, $T/\mu = 0.1$ and $G/\mu = 0.1$. (Right) Zoom on the Brillouin zone boundary at $k = G/2$ (region indicated by a black frame from on the left-hand plot) showing the level repulsion and the gapped sound mode at the edge of the zone. The gray dots are the hydrodynamic prediction given by numerically finding the roots of the determinant of Eq. (43).

this higher Bloch mode mixes with the long distance late time $k = 0$ sound modes.

We showed how one can compute the explicit pattern and strengths of these mixings from the underlying hydrodynamics. As is standard but ever so useful in hydrodynamics is that the behavior of both the patterns and the strengths can be expressed in underlying thermodynamic quantities, notably the susceptibilities, combined with the transport coefficients.

An important feature of a periodic modulation — well known in the condensed matter physics context — is that it breaks translational symmetry. For a perturbatively small lattice the correction to the momentum pole can be interpreted as the momentum relaxation rate and our result agrees with the relaxation rate obtained through the memory matrix formalism, as it should. It is important to emphasize once more that even though there is one relaxation rate, this relaxation rate has two contributions $\Gamma_d$ and $\Gamma_\eta$ corresponding to the two longitudinal diffusive processes. A priori these can have different scaling in temperature.[14] They also exhibit different scaling in the lattice wavevector $G$. Due to this, in systems with charge disorder parametrized as an averaging over many independent lattices, one of these terms will dominate. It is rather the other aspect of the periodic modulation — the presence of Bloch modes in higher Brillouin zones that we wish to emphasize here. At finite density this includes an Umklapped charge diffusion mode. As we analyzed in a companion paper, this mode may be of relevance in condensed matter physics [10]. The strange metal phase of high $T_c$ superconductors shows the development of a mysterious mid IR peak in the optical conductivity at temperatures $T \simeq 300K$; see e.g. [39, 40]. The phenomenology of this peak is almost exactly reproduced by a collision between the Drude pole and the Umklapped charge diffusion pole in a holographic model of the strange metal dual to the Gubser Rocha black hole [10]. If it can be experimentally verified that charge transport in the strange metal is in fact hydrodynamical, this will be the explanation of that phenomenon.

Finally, we verified our results by numerically computing response functions in strongly coupled systems holographically dual to Reissner-Nordström and Gubser-Rocha black holes. The important feature is that hydrodynamics emerges naturally in holographic systems and is not an input. In the computed optical conductivities, we found precisely the poles matching those predicted by our hydrodynamics computation. As as function of varying parameters such as momentum and lattice strength, these poles show complicated behavior including pole collisions and level repulsion denoting various regime changes.

We conclude with emphasizing that the hydrodynamics description of those holographic systems remains valid throughout these collisions and level repulsions. This contrasts with recent studies on the validity of hydrodynamics postulated as a pole collision/level repulsion with a first UV (gapped) pole [41–43]. Our result here shows that this identification has to be done with care. The Umklapped modes are also a priori gapped modes in the zero momentum limit $k \to 0$. However, they remain modes of the conserved charges, can be fully captured in a hydrodynamic description and play a different role from non-hydrodynamic UV modes.

The analysis carried in this paper crucially relied on a static background charge distribution to mimic the effects of a frozen ionic lattice. This ignores the effect of lattice vibrations. Including phonon modes would require a different setup. Moreover, the assumption of local thermal equilibrium rather strongly constrains the hierarchy of scales as $\omega, k \ll G \ll T$. While the results we have achieved are rather general and only rely on the presence of global symmetries and periodicity – which would seem to imply this is valid for a wide range of metallic systems – one must remain cautious as to whether such

---

[14]This is the case in RN where $\Gamma_\eta \sim T^0$ and $\Gamma_d \sim T^2$ while in GR, they both scale with temperature with $\Gamma_\eta \sim \Gamma_d \sim T$.

hierarchy of scales is realized within physical systems.

# Acknowledgements

We are very grateful to F. Balm and J. Zaanen for collaboration in the early stages of this project. We also thank D. Brattan, B. Goutéraux, K. Grosvenor, V. Ziogas and especially A. Krikun for discussions during the Nordita scientific program "Recent developments in strongly correlated quantum matter". This research was supported in part by the FOM program 167 (*Strange Metals*), by the Dutch Research Council (NWO) project 680-91-116 (*Planckian Dissipation and Quantum Thermalisation: From Black Hole Answers to Strange Metal Questions.*), and by the Dutch Research Council/Ministry of Education. The numerical computations were carried out on the Dutch national Cartesius and Snellius national supercomputing facilities with the support of the SURF Cooperative as well as on the ALICE-cluster of Leiden University. We are grateful for their help.

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

## A   Thermodynamics and susceptibilities

In this section of the supplementary material, we will briefly review some key thermodynamic identities related to the static susceptibilities. We will be interested in the conserved charges $\{\delta\epsilon, \delta n\}$ and their associated sources $\{\delta\lambda_\epsilon, \delta\lambda_n\} = \{\delta T/T, \delta\mu - \frac{\mu}{T}\delta T\}$. Since we will be focusing on thermodynamics, we will only be interested in the equilibrium solution and therefore we will drop the $\bar{X}$ notation for background thermodynamics quantities. As a reminder, the susceptibility matrix in the $(\epsilon, n)$ sector is defined as

$$\begin{pmatrix} \delta\epsilon \\ \delta n \end{pmatrix} = \chi \cdot \begin{pmatrix} \delta\lambda_\epsilon \\ \delta\lambda_n \end{pmatrix} , \quad \chi = \begin{pmatrix} \chi_{\epsilon\epsilon} & \chi_{\epsilon n} \\ \chi_{n\epsilon} & \chi_{nn} \end{pmatrix} = \begin{pmatrix} T\left(\dfrac{\partial\epsilon}{\partial T}\right)_{\mu/T} & \dfrac{1}{T}\left(\dfrac{\partial\epsilon}{\partial\mu/T}\right)_T \\ T\left(\dfrac{\partial n}{\partial T}\right)_{\mu/T} & \dfrac{1}{T}\left(\dfrac{\partial n}{\partial\mu/T}\right)_T \end{pmatrix} , \quad (62)$$

while the momentum susceptibility is $\chi_{\pi\pi} = \epsilon + P$. Furthermore, we have the thermodynamic identity

$$T\mathrm{d}X = T\left(\frac{\partial X}{\partial T}\right)_{\mu/T} \mathrm{d}T + T\left(\frac{\partial X}{\partial\mu/T}\right)_T \mathrm{d}(\mu/T) , \tag{63a}$$

$$= \left[T\left(\frac{\partial X}{\partial T}\right)_{\mu/T} - \mu\left(\frac{\partial X}{\partial\mu}\right)_T\right]\mathrm{d}T + T\left(\frac{\partial X}{\partial\mu}\right)_T \mathrm{d}\mu , \tag{63b}$$

$$= T\left(\frac{\partial X}{\partial T}\right)_\mu \mathrm{d}T + T\left(\frac{\partial X}{\partial\mu}\right)_T \mathrm{d}\mu , \tag{63c}$$

such that $T\left(\dfrac{\partial X}{\partial T}\right)_{\mu/T} = T\left(\dfrac{\partial X}{\partial T}\right)_\mu + \mu\left(\dfrac{\partial X}{\partial\mu}\right)_T$. Using this relation and the first law $\mathrm{d}\epsilon = T\mathrm{d}s + \mu\mathrm{d}n$, we have

$$\chi_{\epsilon n} = \left(\frac{\partial\epsilon}{\partial\mu}\right)_T = T\left(\frac{\partial s}{\partial\mu}\right)_T + \mu\left(\frac{\partial n}{\partial\mu}\right)_T \tag{64a}$$

$$= T\frac{\mathrm{d}^2 P}{\mathrm{d}\mu\mathrm{d}T} + \mu\left(\frac{\partial n}{\partial\mu}\right)_T = T\left(\frac{\partial n}{\partial T}\right)_\mu + \mu\left(\frac{\partial n}{\partial\mu}\right)_T = T\left(\frac{\partial n}{\partial T}\right)_{\mu/T} . \tag{64b}$$

Looking back at (62), this means that $\chi_{n\epsilon} = \chi_{\epsilon n}$.

When considering conformal matter in Sec. 4, we used that $P = \epsilon/d$ which is directly implied by the equation of state (58). In that particular case,

$$\chi_{n\epsilon} = \chi_{\epsilon n} = \left(\frac{\partial \epsilon}{\partial \mu}\right)_T = \left(\frac{\partial P}{\partial \mu}\right)_T d = nd \ , \tag{65}$$

$$\chi_{\epsilon\epsilon} = T\left(\frac{\partial \epsilon}{\partial T}\right)_{\mu/T} = d\left(T\left(\frac{\partial P}{\partial T}\right)_\mu + \mu\left(\frac{\partial P}{\partial \mu}\right)_T\right) = sT + \mu n = (\epsilon + P)d \ . \tag{66}$$

## B  Onsager relations

One of the important checks we must make that our dynamical system is well-defined is that it respects Onsager's relations. These can be derived by considering how the system behaves under time-reversal invariance. Given the anti-unitary operator $T$ such that $[H, T] = 0$, we can classify each of the operators associated to our hydrodynamical variables by their representation under this operator. For a given operator $\delta X_a$, we will have $T\delta X_a(t, x)T^{-1} = \eta_a \delta X_a(-t, x)$ with $\eta_a = \pm 1$. Denoting the retarded Green's function associated to a dynamical matrix $K_{ab}$ by $G^R_{ab}$, we have

$$G^R_{ab}(t - t', x, x') = -i\Theta(\tau) \operatorname{tr}\left(\rho[\delta X_a(\tau, x), \delta X_b(0, x')]\right) \ , \qquad \text{with } \rho = e^{-\beta H}/Z \ . \tag{67}$$

We can then see that, due to the anti-unitarity of $T$,

$$G^R_{ab}(\tau, x, x') = \eta_a \eta_b G^R_{ba}(\tau, x', x) \ . \tag{68}$$

In Fourier space, this means $\hat{G}^R_{ab}(\omega, p, p') = \eta_a \hat{G}^R_{ba}(\omega, -p', -p)\eta_b$ and specifically for our periodic background, we can write the Green's function as [22]

$$\hat{G}^R_{ab}(\omega, p, p') = \hat{G}^{R(n,m)}_{ab}(\omega, k) \ , \quad p = k + nG \ , \quad p' = -k + mG \ , \tag{69}$$

where we have used that the discrete lattice symmetry $G^R_{ab}(\tau, x, x') = G^R_{ab}(\tau, x + \frac{2\pi}{G}, x' + \frac{2\pi}{G})$ implies that $p + p' \in \mathbb{Z}G$. For this decomposition, the Onsager relation becomes

$$\hat{G}^{R(n,m)}_{ab}(\omega, k) = \eta_a \hat{G}^{R(-m,-n)}_{ba}(\omega, -k)\eta_b \tag{70a}$$

$$= \eta_a (\hat{G}^\intercal)^{R(-n,-m)}_{ab}(\omega, -k)\eta_b \tag{70b}$$

$$= S \cdot (\hat{G}^\intercal)^{R(-n,-m)}(\omega, -k) \cdot S \ , \tag{70c}$$

$$\hat{G}^R(\omega, k) = S \cdot \overline{(\hat{G}^\intercal)^R(\omega, k)} \cdot S \ . \tag{70d}$$

In the previous expression, we have introduced $S$ the diagonal matrix of eigenvalues $\eta$ and the notation $\overline{\hat{G}^{R(n,m)}(\omega, k)} \equiv \hat{G}^{R(-n,-m)}(\omega, -k)$. It is easy to check that for our background, $\overline{\chi} = \chi$ and therefore since $\chi = \hat{G}^R(\omega = 0)$, we also have $\chi = S \cdot \chi^\intercal \cdot S^{-1}$. We can write this relation in terms of the matrix of couplings

$$N(k) \equiv M(k) \cdot \chi = (i\omega + \hat{K}(\omega, k)) \cdot \chi = i\omega\chi + \hat{\mathcal{K}}(\omega, k) \ , \tag{71}$$

The elements of $N$ are simply the coefficients of the equations (6) written in terms of the sources and expanded in the basis (34). By using that $\hat{G}^R = (1 + i\omega K^{-1}) \cdot \chi$, the relation (70) can then be written as

$$N = S \cdot \overline{N^\intercal} \cdot S^{-1} \ . \tag{72}$$

Let us apply this to the conservation equations (12) for an in-going momentum $p$ and an outgoing momentum $p'$

$$N_{\epsilon n}(p, p') = \sigma_Q \int \mathrm{d}q \, (-pq\bar{\mu}(q))\delta(p + q - p') \, , \tag{73a}$$

$$N_{n\epsilon}(p, p') = \sigma_Q \int \mathrm{d}q \, (pq\bar{\mu}(q) + q^2\bar{\mu}(q))\delta(p + q - p') \, , \tag{73b}$$

$$N_{\epsilon\pi}(p, p') = - \int \mathrm{d}q \, (p\chi_{\pi\pi,0}(q) + q\bar{\epsilon}(q))\delta(p + q - p') \, , \tag{73c}$$

$$N_{\pi\epsilon}(p, p') = \int \mathrm{d}q \, (-p'\chi_{\pi\pi,0}(q) + q\bar{\epsilon}(q))\delta(p + q - p') \, , \tag{73d}$$

$$N_{n\pi}(p, p') = \int \mathrm{d}q \, (-p'\bar{n}(q))\delta(p + q - p') \, , \tag{73e}$$

$$N_{\pi n}(p, p') = \int \mathrm{d}q \, (-p\bar{n}(q))\delta(p + q - p') \, . \tag{73f}$$

We now want to check the Onsager condition for $N$ using that $\eta_\epsilon = \eta_n = -\eta_\pi = 1$. This can be done as follows for the $(\epsilon, n)$ sub-sector

$$N_{n\epsilon}(p', p) = \sigma_Q \int \mathrm{d}q \, (p'q\bar{\mu}(q) + q^2\bar{\mu}(q))\delta(p' + q - p) \, , \tag{74a}$$

$$= \sigma_Q \int \mathrm{d}q \, ((p - q)q\bar{\mu}(q) + q^2\bar{\mu}(q))\delta(p' + q - p) \, , \tag{74b}$$

$$= \sigma_Q \int \mathrm{d}q \, (pq\bar{\mu}(q))\delta(p' + q - p) \, , \tag{74c}$$

$$\eta_\epsilon \eta_n \overline{N_{n\epsilon}(p', p)} = -\sigma_Q \int \mathrm{d}q \, (pq\bar{\mu}(q))\delta(-p' + q + p) = N_{\epsilon n}(p, p') \, , \tag{74d}$$

while for the momentum-charge sector, we have

$$N_{n\pi}(p', p) = \int \mathrm{d}q \, (-p\bar{n}(q))\delta(p' + q - p) \, , \tag{75a}$$

$$\eta_n \eta_\pi \overline{N_{n\pi}(p', p)} = - \int \mathrm{d}q \, (p\bar{n}(q))\delta(-p' + q + p) = N_{\pi n}(p, p') \, . \tag{75b}$$

Finally, we only have to check the energy-momentum sector

$$N_{\epsilon\pi}(p', p) = - \int \mathrm{d}q \, (p'\chi_{\pi\pi,0}(q) + q\bar{\epsilon}(q))\delta(p' + q - p) \, , \tag{76a}$$

$$\eta_\epsilon \eta_\pi \overline{N_{\epsilon\pi}(p', p)} = \int \mathrm{d}q \, (-p'\chi_{\pi\pi,0}(q) + q\bar{\epsilon}(q))\delta(-p' + q + p) = N_{\pi\epsilon}(p, p') \, . \tag{76b}$$

We see therefore that the Onsager reciprocal relations are obeyed by our equations (12).

## C    Second order corrections in lattice strength

Let us consider here a dynamical matrix $\hat{\mathcal{K}}$ of size $N \times N$. The modes of this matrix are given by the solutions to the polynomial equation $\mathcal{P}(\omega) \equiv \det \hat{\mathcal{K}} = 0$, where we can write

$$\mathcal{P}(\omega) = \sum_{n=0}^{N} a_n \omega^n \, . \tag{77}$$

Suppose the coefficients $a_n = \sum_p a_{n,p} A^p$ have a power series expansion in a parameter $A$. We are now interested in perturbative solutions $\omega = \bar{\omega} + A^2 \omega_2$, around a given $A = 0$ solution $\mathcal{P}(\bar{\omega})|_{A=0} = 0$. To do so we can define auxiliary polynomials $\mathcal{P}_p(\omega) = \sum_{n=0}^{N} a_{n,p} \omega^n$ such that

$$\mathcal{P}(\omega) = \sum_p A^p \mathcal{P}_p(\omega) \ . \tag{78}$$

We can now expand the equation $\mathcal{P}(\omega) = 0$ in $A$ at leading and subleading orders, and we find the following two conditions

$$\mathcal{P}_0(\bar{\omega}) = 0 \ , \qquad \omega_2 = -\frac{\mathcal{P}_2(\bar{\omega})}{\mathcal{P}_0'(\bar{\omega})} \ . \tag{79}$$

The first equation is simply the leading order of the mode when there is no lattice while the second equation gives us the subleading correction. Finally, all the coefficients $a_{n,p}$ are themselves polynomials in $G, k$ which can be further expanded in order to get the corrections at higher order in momentum.[15]

So far, it was implicitly assumed that the modes have no degeneracy when $A \to 0$ as that would imply that $\mathcal{P}_0'(\bar{\omega}) = 0$. When that happens, the correction is given by higher order terms with

$$\mathcal{P}_0(\bar{\omega}) = 0 \ , \qquad \mathcal{P}_2(\bar{\omega}) = 0 \ , \qquad \omega_2 = \frac{-\mathcal{P}_2'(\bar{\omega}) \pm \sqrt{(\mathcal{P}_2'(\bar{\omega}))^2 - 4\mathcal{P}_4(\bar{\omega})\mathcal{P}_0''(\bar{\omega})}}{2\mathcal{P}_0''(\bar{\omega})} \ . \tag{80}$$

This is the case for $\hat{\mathcal{K}}_{L+}$ with $\bar{\omega} = 0$. However, it turns out that for this matrix, $\mathcal{P}_4(\bar{\omega}) = 0$ and $\mathcal{P}_2'(\omega) = 0$, so the two degenerate poles remain degenerate at this order in perturbation theory, with $\omega_2 = 0$. A simpler way to see this is also to notice that the first line of $\hat{\mathcal{K}}_{L+}$ is proportional to $i\omega$ and therefore so will be $\det \hat{\mathcal{K}}_{L+}$. Consequently, this sector admits an exact conservation mode and we can use our non-degenerate method on the $4 \times 4$ lower-right sub-block of this matrix where there is no degeneracy left. We would then see that the other $\bar{\omega} = 0$ pole also remains unshifted $\omega_2 = 0$.

All that is therefore needed to compute the corrections in (54) and (57) is to know the

---

[15]Note that in this method, the order of limits is chosen such that at finite $k$, we would be getting the $k > AV_{\text{int}}$ branch of solutions.

coefficients $a_{n,p}$ for a given matrix $\hat{\mathcal{K}}$. In the case of $\hat{\mathcal{K}}_{L-}$, we have

$$a_{0,0} = 0 \;, \quad a_{1,0} = -i\left(\chi_{\pi\pi,0}\right)^3 \sigma_Q G^4 \;, \quad a_{4,0} = \left(\chi_{\pi\pi,0}\right)^2 d_\chi \;, \tag{81a}$$

$$a_{2,0} = -\chi_{\pi\pi,0}\left[\chi_{\epsilon\epsilon,0}n_0^2 - 2\chi_{n\epsilon,0}\chi_{\pi\pi,0}n_0 + \chi_{nn,0}\left(\chi_{\pi\pi,0}\right)^2\right]G^2 - \chi_{\epsilon\epsilon,0}\chi_{\pi\pi,0}\hat{\eta}\sigma_Q G^4 \;, \tag{81b}$$

$$a_{3,0} = i\chi_{\pi\pi,0}\left(d_\chi\hat{\eta} + \chi_{\epsilon\epsilon,0}\chi_{\pi\pi,0}\sigma_Q\right)G^2 \;, \tag{81c}$$

$$a_{0,2} = \frac{\mu_0^2}{2}\left(\chi_{n\epsilon,0}n_0 - \chi_{nn,0}\chi_{\pi\pi,0}\right)^2 G^4 + \frac{\mu_0^2}{2}\left(\chi_{n\epsilon,0}\right)^2\hat{\eta}\sigma_Q G^6 \;, \tag{81d}$$

$$a_{1,2} = -i\frac{1}{2}G^4\left[\chi_{nn,0}\hat{\eta}\mu_0^2 d_\chi + \chi_{\pi\pi,0}\sigma_Q\left(2\mu_0(\chi_{n\epsilon,1}^{(1)}\chi_{\pi\pi,0} - 2\chi_{n\epsilon,0}\chi_{\pi\pi,1}^{(1)}) + \mu_0^2\left(\chi_{n\epsilon,0}^2 + \chi_{nn,0}\chi_{\pi\pi,0}\right)\right.\right. \tag{81e}$$

$$\left.\left. + 2\chi_{\pi\pi,2}^{(0)}\chi_{\pi\pi,0} + \mu_0^2 n_0^2\right)\right] - i\frac{\mu_0^2}{2}\chi_{\pi\pi,0}\hat{\eta}\sigma_Q^2 G^6 \;, \tag{81f}$$

$$a_{2,2} = \frac{1}{2}G^2\left[-2n_0^2(\chi_{\epsilon\epsilon,2}^{(0)}\chi_{\pi\pi,0} + \chi_{\epsilon\epsilon,0}\chi_{\pi\pi,2}^{(0)}) + \chi_{\pi\pi,0}n_0(\mu_0(-2\chi_{\epsilon\epsilon,0}\chi_{nn,1}^{(1)} + 2\chi_{n\epsilon,1}^{(1)}\chi_{n\epsilon,0} + \chi_{n\epsilon,0}\chi_{nn,0}\mu_0)\right. \tag{81g}$$

$$\left. + 4\chi_{n\epsilon,2}^{(0)}\chi_{\pi\pi,0} + 4\chi_{n\epsilon,0}\chi_{\pi\pi,2}^{(0)}) - \chi_{\pi\pi,0}\left(\chi_{nn,0}\mu_0^2\left(\chi_{nn,0}\chi_{\pi\pi,0} + d_\chi\right)\right.\right. \tag{81h}$$

$$\left.\left. + 2\chi_{\pi\pi,0}\mu_0(\chi_{n\epsilon,1}^{(1)}\chi_{nn,0} - \chi_{n\epsilon,0}\chi_{nn,1}^{(1)}) + 2\chi_{\pi\pi,0}(\chi_{nn,2}^{(0)}\chi_{\pi\pi,0} + \chi_{nn,0}\chi_{\pi\pi,2}^{(0)})\right) + 4\chi_{\pi\pi,1}^{(1)}\mu_0 n_0 d_\chi\right] \tag{81i}$$

$$- \frac{\sigma_Q}{2}G^4\left[\chi_{\pi\pi,0}\left(2\chi_{\epsilon\epsilon,2}^{(0)}\hat{\eta} + \chi_{nn,0}\hat{\eta}\mu_0^2 + \chi_{\pi\pi,0}\mu_0^2\sigma_Q\right) + 2\chi_{\epsilon\epsilon,0}\chi_{\pi\pi,2}^{(0)}\hat{\eta}\right] \;, \tag{81j}$$

$$a_{3,2} = \frac{1}{2}iG^2\left[2\hat{\eta}\left(\chi_{\epsilon\epsilon,2}^{(0)}\chi_{nn,0}\chi_{\pi\pi,0} + \chi_{\epsilon\epsilon,0}\chi_{nn,2}^{(0)}\chi_{\pi\pi,0} + d_\chi\chi_{\pi\pi,2}^{(0)} - 2\chi_{n\epsilon,2}^{(0)}\chi_{n\epsilon,0}\chi_{\pi\pi,0}\right)\right. \tag{81k}$$

$$\left. + \chi_{\pi\pi,0}\sigma_Q\left(2\chi_{\epsilon\epsilon,2}^{(0)}\chi_{\pi\pi,0} + 4\chi_{\epsilon\epsilon,0}\chi_{\pi\pi,2}^{(0)} + \chi_{nn,0}\chi_{\pi\pi,0}\mu_0^2\right) - 4\chi_{\epsilon\epsilon,0}(\chi_{\pi\pi,1}^{(1)})^2\sigma_Q\right] \;, \tag{81l}$$

$$a_{4,2} = \chi_{\pi\pi,0}\left[\chi_{\epsilon\epsilon,2}^{(0)}\chi_{nn,0}\chi_{\pi\pi,0} + \chi_{\epsilon\epsilon,0}\chi_{nn,2}^{(0)}\chi_{\pi\pi,0} + 2d_\chi\chi_{\pi\pi,2}^{(0)} - 2\chi_{n\epsilon,2}^{(0)}\chi_{n\epsilon,0}\chi_{\pi\pi,0}\right] - 2d_\chi(\chi_{\pi\pi,1}^{(1)})^2 \;. \tag{81m}$$

On the other hand, for the $4 \times 4$ lower-right sub-block of the matrix $\hat{\mathcal{K}}_{L+}$, we find the

following coefficients for the determinant

$$a_{0,0} = 0 \ , \qquad a_{1,0} = -i\chi_{\epsilon\epsilon,0}(\chi_{\pi\pi,0})^2\sigma_Q G^4 \ , \qquad a_{4,0} = \chi_{\pi\pi,0}\chi_{\epsilon\epsilon,0}d_\chi \ , \tag{82a}$$

$$a_{2,0} = -\chi_{\epsilon\epsilon,0}\left[\chi_{\epsilon\epsilon,0}n_0^2 - 2\chi_{n\epsilon,0}\chi_{\pi\pi,0}n_0 + \chi_{nn,0}(\chi_{\pi\pi,0})^2\right]G^2 - (\chi_{\epsilon\epsilon,0})^2\hat{\eta}\sigma_Q G^4 \ , \tag{82b}$$

$$a_{3,0} = i\chi_{\epsilon\epsilon,0}G^2\left(d_\chi\hat{\eta} + \chi_{\epsilon\epsilon,0}\chi_{\pi\pi,0}\sigma_Q\right) \ , \tag{82c}$$

$$a_{0,2} = 0 \ , \tag{82d}$$

$$a_{1,2} = -\frac{1}{2}iG^4\sigma_Q\left[\chi_{\pi\pi,0}\left(2\chi_{\epsilon\epsilon,2}^{(0)}\chi_{\pi\pi,0} + 2\chi_{n\epsilon,1}^{(1)}\mu_0(\chi_{\epsilon\epsilon,0} - 2\chi_{\pi\pi,0}) + \chi_{nn,0}\mu_0^2(\chi_{\epsilon\epsilon,0} + \chi_{\pi\pi,0})\right) + \chi_{\epsilon\epsilon,0}\mu_0^2 n_0^2\right] \tag{82e}$$

$$-\frac{1}{2}i\chi_{\epsilon\epsilon,0}\hat{\eta}G^6\mu_0^2\sigma_Q^2 \ , \tag{82f}$$

$$a_{2,2} = \frac{1}{2}G^2\left[n_0^2\left(-4\chi_{\epsilon\epsilon,2}^{(0)}\chi_{\epsilon\epsilon,0} + 4(\chi_{\epsilon\epsilon,1}^{(1)})^2 + 4\mu_0(\chi_{\epsilon\epsilon,0}\chi_{n\epsilon,1}^{(1)} - \chi_{\epsilon\epsilon,1}^{(1)}\chi_{n\epsilon,0}) - d_\chi\mu_0^2\right)\right. \tag{82g}$$

$$+4\chi_{\pi\pi,0}n_0(\chi_{\epsilon\epsilon,2}^{(0)}\chi_{n\epsilon,0} - 2\chi_{\epsilon\epsilon,1}^{(1)}\chi_{n\epsilon,1}^{(1)} + \chi_{\epsilon\epsilon,1}^{(1)}\chi_{nn,0}\mu_0 + \chi_{\epsilon\epsilon,0}\chi_{n\epsilon,2}^{(0)} - \chi_{n\epsilon,1}^{(1)}\chi_{n\epsilon,0}\mu_0) \tag{82h}$$

$$-\chi_{\pi\pi,0}\left(2\chi_{\pi\pi,0}\left(\chi_{\epsilon\epsilon,2}^{(0)}\chi_{nn,0} + \chi_{\epsilon\epsilon,0}\chi_{nn,2}^{(0)} - 2(\chi_{n\epsilon,1}^{(1)})^2\right) + 2\chi_{\epsilon\epsilon,0}\mu_0(\chi_{n\epsilon,1}^{(1)}\chi_{nn,0} - \chi_{n\epsilon,0}\chi_{nn,1}^{(1)})\right. \tag{82i}$$

$$\left.+\chi_{\epsilon\epsilon,0}(\chi_{nn,0})^2\mu_0^2\right) + \chi_{\epsilon\epsilon,0}\mu_0 n_0(-2\chi_{\epsilon\epsilon,0}\chi_{nn,1}^{(1)} + 2\chi_{n\epsilon,1}^{(1)}\chi_{n\epsilon,0} + \chi_{n\epsilon,0}\chi_{nn,0}\mu_0)\right] \tag{82j}$$

$$+\frac{1}{2}G^4\sigma_Q\left[\hat{\eta}\left(-4\chi_{\epsilon\epsilon,2}^{(0)}\chi_{\epsilon\epsilon,0} + 4(\chi_{\epsilon\epsilon,1}^{(1)})^2 + 4\mu_0(\chi_{\epsilon\epsilon,0}\chi_{n\epsilon,1}^{(1)} - \chi_{\epsilon\epsilon,1}^{(1)}\chi_{n\epsilon,0}) + \mu_0^2\left((\chi_{n\epsilon,0})^2 - 2\chi_{\epsilon\epsilon,0}\chi_{nn,0}\right)\right)\right. \tag{82k}$$

$$\left.-\chi_{\epsilon\epsilon,0}\chi_{\pi\pi,0}\mu_0^2\sigma_Q\right] \ , \tag{82l}$$

$$a_{3,2} = \frac{1}{2}iG^2\left[-2\hat{\eta}\left(-\chi_{\epsilon\epsilon,0}\left(2\chi_{\epsilon\epsilon,2}^{(0)}\chi_{nn,0} + \chi_{\epsilon\epsilon,0}\chi_{nn,2}^{(0)} - 2(\chi_{n\epsilon,1}^{(1)})^2\right) + \chi_{\epsilon\epsilon,2}^{(0)}(\chi_{n\epsilon,0})^2\right.\right. \tag{82m}$$

$$\left.\left.+2(\chi_{\epsilon\epsilon,1}^{(1)})^2\chi_{nn,0} + 2\chi_{n\epsilon,0}(\chi_{\epsilon\epsilon,0}\chi_{n\epsilon,2}^{(0)} - 2\chi_{\epsilon\epsilon,1}^{(1)}\chi_{n\epsilon,1}^{(1)})\right)\right. \tag{82n}$$

$$+\chi_{\pi\pi,0}\sigma_Q\left(4\chi_{\epsilon\epsilon,2}^{(0)}\chi_{\epsilon\epsilon,0} - 4(\chi_{\epsilon\epsilon,1}^{(1)})^2 + \mu_0(4\chi_{\epsilon\epsilon,1}^{(1)}\chi_{n\epsilon,0} - 4\chi_{\epsilon\epsilon,0}\chi_{n\epsilon,1}^{(1)}) - \mu_0^2\left(\chi_{n\epsilon,0}^2 - 2\chi_{\epsilon\epsilon,0}\chi_{nn,0}\right)\right) \tag{82o}$$

$$\left.+2(\chi_{\epsilon\epsilon,0})^2\chi_{\pi\pi,2}^{(0)}\sigma_Q\right] \ , \tag{82p}$$

$$a_{4,2} = \chi_{\pi\pi,0}\left[2\chi_{\epsilon\epsilon,2}^{(0)}\chi_{\epsilon\epsilon,0}\chi_{nn,0} - \chi_{\epsilon\epsilon,2}^{(0)}(\chi_{n\epsilon,0})^2 - 2(\chi_{\epsilon\epsilon,1}^{(1)})^2\chi_{nn,0} + 4\chi_{\epsilon\epsilon,1}^{(1)}\chi_{n\epsilon,1}^{(1)}\chi_{n\epsilon,0}\right. \tag{82q}$$

$$\left.+(\chi_{\epsilon\epsilon,0})^2\chi_{nn,2}^{(0)} - 2\chi_{\epsilon\epsilon,0}\chi_{n\epsilon,2}^{(0)}\chi_{n\epsilon,0} - 2\chi_{\epsilon\epsilon,0}(\chi_{n\epsilon,1}^{(1)})^2\right] + \chi_{\epsilon\epsilon,0}\chi_{\pi\pi,2}^{(0)}d_\chi \tag{82r}$$

The coefficients for the determinant of the full longitudinal matrix (45) are too involved to be written down here but can be obtained in the exact same way. All these corrections were derived using Mathematica.

# D  Numerical computations in strongly coupled field theories dual to Reissner-Nordström and Gubser-Rocha AdS black holes: set-up

## D.1  Thermodynamics

In Sec. 4, we focused on the specific conformal hydrodynamics that emerges at long wavelength and low frequencies from the holographic dynamics of the RN and GR black holes. In equilibrium the thermodynamic equation of state of each is given by (59) and (60) re-

spectively. From these, we can determine the charge density $n_0 = \left( \dfrac{\partial P}{\partial \mu} \right)_T$ as well as the entropy density $s_0 = \left( \dfrac{\partial P}{\partial T} \right)_\mu$ while the energy density just follows from conformal invariance and is given by $\epsilon_0 = 2P_0$. One can further compute the various susceptibilities $\chi^{(n)}_{ab,m}$ appearing in the hydrodynamics expressions of Sec. 3. As a reminder from Appendix A, the conformal equation of state also imposes $\chi_{\epsilon\epsilon,0} = 2\chi_{\pi\pi,0}$ and $\chi_{n\epsilon,0} = \chi_{\epsilon n,0} = 2n_0$. For the RN black hole, the various susceptibilities and thermodynamic quantities are

$$n_0 = \frac{\mu_0}{6}\sqrt{3\mu_0^2 + 16\pi^2 T_0^2} + \frac{2\pi\mu_0 T_0}{3} = \frac{\chi_{n\epsilon,0}}{2} = \frac{\chi_{\epsilon n,0}}{2} \ , \tag{83a}$$

$$s_0 = \frac{8\pi^2}{9}T_0\sqrt{3\mu_0^2 + 16\pi^2 T_0^2} + \frac{\pi}{9}\left(3\mu_0^2 + 32\pi^2 T_0^2\right) \ , \tag{83b}$$

$$\chi_{nn,0} = \frac{1}{6}\left( \frac{2\left(3\mu_0^2 + 8\pi^2 T_0^2\right)}{\sqrt{3\mu_0^2 + 16\pi^2 T_0^2}} + 4\pi T_0 \right) \ , \tag{83c}$$

$$\chi_{\pi\pi,0} = \frac{3\mu_0^4\left(2\pi T_0\left(4\pi T_0 - \sqrt{3\mu_0^2 + 16\pi^2 T_0^2}\right) + \mu_0^2\right)}{2\left(\sqrt{3\mu_0^2 + 16\pi^2 T_0^2} - 4\pi T_0\right)^3} = \frac{\chi_{\epsilon\epsilon,0}}{2} \ , \tag{83d}$$

$$\chi_{nn,1}^{(1)} = \frac{3\mu_0^2\left(\mu_0^2 + 8\pi^2 T_0^2\right)}{2\left(3\mu_0^2 + 16\pi^2 T_0^2\right)^{3/2}} \ , \qquad \chi_{n\epsilon,1}^{(1)} = \frac{\mu_0}{6}\left( \frac{2\left(3\mu_0^2 + 8\pi^2 T_0^2\right)}{\sqrt{3\mu_0^2 + 16\pi^2 T_0^2}} + 4\pi T_0 \right) \ , \tag{83e}$$

$$\chi_{\pi\pi,1}^{(1)} = \frac{\mu_0^2}{4}\sqrt{3\mu_0^2 + 16\pi^2 T_0^2} + \pi\mu_0^2 T_0 = \frac{\chi_{\epsilon\epsilon,1}^{(1)}}{2} \ , \tag{83f}$$

$$\chi_{nn,2}^{(0)} = \frac{96\pi^4\mu_0^2 T_0^4}{\left(3\mu_0^2 + 16\pi^2 T_0^2\right)^{5/2}} \ , \qquad \chi_{n\epsilon,2}^{(0)} = \frac{3\mu_0^3\left(\mu_0^2 + 8\pi^2 T_0^2\right)}{2\left(3\mu_0^2 + 16\pi^2 T_0^2\right)^{3/2}} \ , \tag{83g}$$

$$\chi_{\pi\pi,2}^{(0)} = \frac{3\mu_0^4 + 2\pi\mu_0^2 T_0\left(\sqrt{3\mu_0^2 + 16\pi^2 T_0^2} + 4\pi T_0\right)}{4\sqrt{3\mu_0^2 + 16\pi^2 T_0^2}} = \frac{\chi_{\epsilon\epsilon,2}^{(0)}}{2} \ , \tag{83h}$$

while for the GR black hole we have

$$n_0 = \frac{\mu_0}{3}\sqrt{3\mu_0^2 + 16\pi^2 T_0^2} = \frac{\chi_{\epsilon n,0}}{2} \ , \qquad s_0 = \frac{16\pi^2}{9}T_0\sqrt{3\mu_0^2 + 16\pi^2 T_0^2} \ , \tag{84a}$$

$$\chi_{nn,0} = \frac{2\left(3\mu_0^2 + 8\pi^2 T_0^2\right)}{3\sqrt{3\mu_0^2 + 16\pi^2 T_0^2}} \ , \qquad \chi_{\pi\pi,0} = \frac{1}{9}\left(3\mu_0^2 + 16\pi^2 T_0^2\right)^{3/2} = \frac{\chi_{\epsilon\epsilon,0}}{2} \ , \tag{84b}$$

$$\chi_{nn,1}^{(1)} = \frac{3\mu_0^2\left(\mu_0^2 + 8\pi^2 T_0^2\right)}{\left(3\mu_0^2 + 16\pi^2 T_0^2\right)^{3/2}} \ , \qquad \chi_{n\epsilon,1}^{(1)} = \frac{2\mu_0\left(3\mu_0^2 + 8\pi^2 T_0^2\right)}{3\sqrt{3\mu_0^2 + 16\pi^2 T_0^2}} \ , \tag{84c}$$

$$\chi_{\pi\pi,1}^{(1)} = \frac{\mu_0^2}{2}\sqrt{3\mu_0^2 + 16\pi^2 T_0^2} = \frac{\chi_{\epsilon\epsilon,1}^{(1)}}{2} \ , \qquad \chi_{nn,2}^{(0)} = \frac{192\pi^4\mu_0^2 T_0^4}{\left(3\mu_0^2 + 16\pi^2 T_0^2\right)^{5/2}} \ , \tag{84d}$$

$$\chi_{n\epsilon,2}^{(0)} = \frac{3\mu_0^3\left(\mu_0^2 + 8\pi^2 T_0^2\right)}{\left(3\mu_0^2 + 16\pi^2 T_0^2\right)^{3/2}} \ , \qquad \chi_{\pi\pi,2}^{(0)} = \frac{\mu_0^2\left(3\mu_0^2 + 8\pi^2 T_0^2\right)}{2\sqrt{3\mu_0^2 + 16\pi^2 T_0^2}} = \frac{\chi_{\epsilon\epsilon,2}^{(0)}}{2} \ . \tag{84e}$$

Lastly, we need to know some information on the transport coefficients $\eta$ and $\sigma_Q$ to compute the hydrodynamic response. These can be determined in the momentum-dependent homogeneous systems through $\eta = \lim_{\omega \to 0} \frac{1}{\omega}\text{Im}\, G_{T_{xy}T_{xy}}(\omega, k = 0)$ and $\sigma_Q = \lim_{\omega \to 0} \frac{1}{\omega}\text{Im}\, G_{J_x J_x}(\omega, k = 0)$. In the case of conformal-to-AdS$_2$ solutions like the RN and GR black holes, these expressions can be solved analytically for the two transport

coefficients. The shear viscosity $\eta$ saturates the minimal viscosity bound $\eta = \frac{s_0}{4\pi}$ [44] while $\sigma_Q$ was computed for a wide class of scaling black hole solutions [45] and here is given by

$$\sigma_Q = \frac{4\pi^2 T_0^2}{9} \left( \frac{\sqrt{3\mu_0^2 + 16\pi^2 T_0^2} - 4\pi T_0}{\mu_0^2 - 2\pi T_0 \sqrt{3\mu_0^2 + 16\pi^2 T_0^2} + 8\pi^2 T_0^2} \right)^2 \quad \text{for RN,} \tag{85a}$$

$$\sigma_Q = \left( 1 + \frac{3\mu_0^2}{16\pi^2 T_0^2} \right)^{-3/2} \quad \text{for GR.} \tag{85b}$$

### D.2   Numerics

We briefly review here how we compute the optical conductivity in the 2+1 dimensional strongly coupled conformal field theory holographically dual to the RN and GR black holes in the presence of a lattice. More details about the numerical methods used to compute these backgrounds and fluctuations can be found in the companion article [10]. The homogeneous RN black hole is a saddle point of the Einstein-Maxwell action

$$S = \int \mathrm{d}^4 x \sqrt{-g} \left[ (R - 2\Lambda) - \frac{1}{4} F_{\mu\nu} F^{\mu\nu} \right] , \tag{86}$$

with metric

$$\mathrm{d}s^2 = g_{\mu\nu} \mathrm{d}x^\mu \mathrm{d}x^\nu = \frac{1}{z^2} \left[ -f(z)\mathrm{d}t^2 + \frac{\mathrm{d}z^2}{f(z)} + \mathrm{d}x^2 + \mathrm{d}y^2 \right] , \qquad A = A_t(z)\mathrm{d}t , \tag{87}$$

where $f(z) = (1-z)\left(1 + z + z^2 - \frac{\mu^2 z^3}{4}\right)$ is the emblackening factor and $A_t(z) = \mu(1-z)$ a U(1) gauge field. In the above expressions, $z$ is the radial coordinate ranging from the AdS boundary at $z = 0$ to the horizon of the black hole at $z = 1$. The temperature of this black hole is $T = \frac{12 - \mu^2}{16\pi}$.[16]

The GR black hole is similarly obtained by extremizing the Einstein-Maxwell-Dilaton action

$$S = \frac{1}{2\kappa^2} \int \mathrm{d}^4 x \sqrt{-g} \left[ R - \frac{Z(\phi)}{4} F_{\mu\nu} F^{\mu\nu} - \frac{1}{2}(\partial_\mu \phi)^2 + V(\phi) \right] , \tag{88}$$

with the potentials $Z(\phi) = e^{\phi/\sqrt{3}}$ and $V(\phi) = 6\cosh(\phi/\sqrt{3})$ Its metric is

$$\mathrm{d}s^2 = g_{\mu\nu} \mathrm{d}x^\mu \mathrm{d}x^\nu = \frac{1}{z^2} \left[ -h(z)\mathrm{d}t^2 + \frac{1}{h(z)}\mathrm{d}z^2 + g(z)(\mathrm{d}x^2 + \mathrm{d}y^2) \right] , \tag{89a}$$

$$A = \sqrt{3Q(1+Q)} \frac{(1-z)}{1+Qz} \mathrm{d}t , \qquad \phi = \frac{\sqrt{3}}{2} \log(1 + Qz) . \tag{89b}$$

The functions $h(z)$ and $g(z)$ are given by

$$h(z) = \frac{(1-z)}{g(z)} \left[ 1 + (1 + 3Q)z + (1 + 3Q(1+Q)) z^2 \right] , \qquad g(z) = (1 + Qz)^{3/2} . \tag{90}$$

This model is similar to the Einstein-Maxwell model with the addition of a neutral scalar field $\phi$ which controls the strength of the $U(1)$ charge through the potential $Z(\phi)$. A

---

[16]A priori, if we allowed the black hole horizon to be arbitrarily located at $z = z_h$, the temperature and chemical potential would be two independent parameters. However, when using the freedom to rescale the radial coordinate such that $z_h = 1$, we have implicitly fixed the temperature as a function of the chemical potential such that the only thermodynamic degree of freedom here is $T/\mu$.

consequence of this is the ability to discharge some of the black hole charge near the horizon such that the extremal $T = 0$ solution of this GR black hole will have a vanishing horizon and therefore vanishing entropy $S_{T=0} = 0$. The distance from extremality is controlled by the parameter $Q$: it is related to the chemical potential through $\mu = \sqrt{3Q(1+Q)}$ and the temperature of the non-extremal black hole is given by $T = \frac{3\sqrt{1+Q}}{4\pi}$.

To obtain backgrounds with an explicit lattice, we will allow for a more general ansatz

$$ds^2 = \frac{1}{z^2}\left(-Q_{tt}f(z)\eta_t^2 + Q_{xx}\eta_x^2 + Q_{yy}\eta_y^2 + \frac{Q_{zz}}{f(z)}\eta_z^2\right), \tag{91a}$$

$$\eta_t = \mathrm{d}t, \qquad \eta_y = \mathrm{d}y, \qquad \eta_z = \mathrm{d}z, \qquad \eta_x = \mathrm{d}x + Q_{xz}\mathrm{d}z, \tag{91b}$$

$$A = \mu(1-z)a_t\mathrm{d}t, \tag{91c}$$

for RN with $f(z)$ unchanged from (87) and

$$ds^2 = \frac{1}{z^2}\left(-Q_{tt}h(z)\eta_t^2 + g(z)\left(Q_{xx}\eta_x^2 + Q_{yy}\eta_y^2\right) + \frac{Q_{zz}}{h(z)}\eta_z^2\right), \tag{92a}$$

$$\eta_t = \mathrm{d}t, \qquad \eta_y = \mathrm{d}y, \qquad \eta_z = \mathrm{d}z, \qquad \eta_x = \mathrm{d}x + Q_{xz}\mathrm{d}z, \tag{92b}$$

$$A = \frac{\mu(1-z)}{1+Qz}a_t\mathrm{d}t, \qquad \phi = \frac{3}{2}\log\left(1 + \varphi(z)Qz\right). \tag{92c}$$

for GR with $h(z)$ and $g(z)$ unchanged from (89), but every field $Q_{ij}, a_t, \varphi$ is now a priori a function of $x$ and $z$. We require these fields to be regular near the horizon[17] and that their UV behaviour at $z = 0$ recovers AdS asymptotics.[18] Moreover, to encode the modulation of the chemical potential, we must impose the following boundary condition on the gauge field

$$a_t(z = 0) = 1 + A\cos(Gx) . \tag{93}$$

The system is then solved numerically for the unknown functions $Q_{ij}, a_t, \varphi$.

To compute the optical conductivity in the holographically dual field theory, we must consider small fluctuations on top of this spatially modulated background. We linearize the Einstein equations around our lattice background

$$g_{\mu\nu} = \bar{g}_{\mu\nu} + \delta h_{\mu\nu}e^{-i\omega t + ikx} , \tag{94a}$$

$$A_\mu = \bar{A}_\mu + \delta b_\mu e^{-i\omega t + ikx} , \tag{94b}$$

$$\varphi = \bar{\varphi} + \delta\psi e^{-i\omega t + ikx} , \tag{94c}$$

and solve for these fluctuations with infalling boundary conditions, corresponding to choosing the response sourced through the retarded Green's function. The response in the radial electric field $F_{zx}$ in answer to an oscillating source in the potential $\delta\partial_t A_x(\omega) \equiv \delta b_x$ keeping the other components sourceless[19] evaluated in the limit $z \to 0$ then translates through the holographic AdS/CFT correspondence into the longitudinal optical conductivity $\sigma = \lim_{z\to 0}\frac{F_{zx}}{\partial_t A_x} = \frac{\delta J_x}{\delta E_x}$ . In the language of our hydrodynamic setup in Sec. 2, this

---

[17]One of the regularity conditions near the horizon is that $Q_{tt}(z = 1) = Q_{zz}(z = 1)$. This choice has the direct consequence that the temperature of the black hole remains constant and given by the homogeneous value for each model.

[18] In the case of the dilaton $\phi$, the UV boundary condition chosen is a multi-trace deformation chosen such that the deformation is marginal and the boundary remains conformal. For more details, see [35].

[19]Note that the condition for the dilaton to be sourceless is non-trivial and inherited from the mixed boundary condition of the background dilaton (see footnote 18).

is akin to simply turning on an external electric field $\delta E_x$ with momentum $k$ and frequency $\omega$. This response is also solved for numerically.

The numerical solutions to these equations were obtained using a publicly available custom package [46] and computed on the Dutch national Cartesius and Snellius super-computers with the support of SURF Cooperative.