# Peer review of "Hydrodynamics of a relativistic charged fluid in the presence of a periodically modulated chemical potential"

_SciPost Physics_

## Round 2 · Referee Report · Anonymous · 2023-8-21

Report

I am happy with the changes to the conclusion made by the authors.

---

## Round 2 · Referee Report · Anonymous · 2023-8-24

Strengths

1 - The developed hydrodynamic framework is novel and the subsequent derived results are backed up by holographic examples.

2 - The relation of the developed framework to other techniques in the literature (e.g. the memory matrix formalism) is outlined.

3 - Novel effects in the dispersion relations of the hydrodynamic modes are discovered and clearly explained.

4 - The work opens up a new pathway in an existing research direction with clear potential for follow up work.

Weaknesses

None other than the requested changes.

Report

The paper is written in a clear and intelligible manner and explains potentially unfamiliar terms as necessary. Similarly, the abstract and introduction clearly explain the context and results of the work.

The appendices describe well how certain results in the paper are produced, and would enable an expert in the field to repeat the numerical analysis used to produce the figures. Similarly, there is sufficient detail in the bulk for a qualified expert to reproduce the presented calculations. Previous works are clearly cited and represent a complete record of the relevant literature.

In my opinion, the analysis is new, interesting and complementary to other approaches in the literature. Subject to satisfying the final requested changes I would recommend publication at SciPost.

Requested changes

I would ask the authors to address the following two points prior to acceptance:

1 - The author's have included the statement "In a non-relativistic system, the static equilibrium is uniquely determined. In a relativistic system -- which we use in this paper -- it is convenient to choose the reference frame for which the fluid is at rest."

This is incorrect, static equilibrium is not uniquely determined for a non-relativistic fluid any more than a relativistic one as Galilean fluids have Galilean boost invariance. I would ask the authors to try and addressing my previous point again.

2 - Unfortunately, I do not agree with the author's previous comment that "the charge density $n(x,t)$ and the chemical potential $\mu(x,t)$ are related by the static charge susceptibility $\chi$ as long as the position $x$ and time $t$ is coarse grained over a region much larger than the local equilibration scale".

This can be readily seen by considering the Martin-Kadanoff procedure. In this case one turns on a chemical potential in the past and adiabatically increases it to some value $\mu_{0}$ at $t=0$. At this time one turns off the chemical potential (set $\mu(0,\vec{x})=0$) and allows the system to evolve hydrodynamically. If, as the authors claim, charge density is always related to the chemical potential by the static susceptibility then the charge density would be zero everywhere. As this is not the case the authors should either clarify or modify their discussion of this point.

  • validity: top
  • significance: high
  • originality: high
  • clarity: high
  • formatting: reasonable
  • grammar: good

Author:  Nicolas Chagnet  on 2023-09-13  [id 3980]

(in reply to Report 2 on 2023-08-24)
Category:
answer to question
reply to objection

Regarding the first point, indeed, the referee is correct in detail. When the thermal equilibrium is homogeneous, the equations of non-relativistic hydrodynamics are invariant under a Galilean boost. There is a subtle issue with charged non-relativistic hydrodynamics in an external chemical potential. Though one might say that the external chemical potential does not transform under Galilean boosts, at some point this comes in conflict with the fact that the electromagnetic sector is not invariant under Galilean boosts, but Lorentz boosts instead, where an external chemical potential transforms into a combination of an external chemical potential plus an external source for the electric current. In that sense an external chemical potential without a current-source does fix a unique frame even in “non-relativistic” hydrodynamics. We fully agree with the referee that the text as written is incorrect, and leaving the subtle issue above undiscussed, we offer to adjust the text accordingly.

The above constitutive relations also hold in a static equilibrium background. In a non-relativistic system, the static equilibrium is uniquely determined. In a relativistic system --- which we use in this paper --- it is convenient to choose the reference frame for which the fluid is at rest.

is to be replaced by

The above constitutive relations also hold in a static equilibrium background. In a system with Galilean or relativistic Lorentz boost invariance --- which we use in this paper --- it is convenient to choose the reference frame for which the equilibrium fluid is at rest.

Regarding the second point, we would like to make our statement more accurate. The Kadanoff-Martin procedure highlighted by the referee is indeed the correct way to think about the response of the system to an external source prepared adiabatically. However, in our previous assertion as well as within the current draft of the manuscript, we were referring to internal conjugate variables. In that context, the internal chemical potential and charge density are not independent variables but are related through the equation of state. This is a requirement in hydrodynamics to close the system of equations. The static susceptibilities relation for fluctuations in the internal chemical potential with respect to the charge density are a consequence of this through the assumption of local thermodynamic equilibrium. Through this assumption, we can assign local values to thermodynamic variables (provided the scales we consider are large enough) and the equation of state is assumed to hold locally $P(t,x) = P[\mu(t,x), T(t,x)]$. The charge density for instance is then simply determined by the local (internal) chemical potential and temperature through $n(t,x) = \frac{\partial P}{\partial \mu(t,x)}[\mu(t,x), T(t,x)]$ and thus local variations of the internal chemical potential $\mu(x,t)$ are related to local variations of $n(t,x)$ through the charge susceptibility $\chi(t,x) = \frac{\partial^2 P}{\partial \mu(x,t)^2}[\mu(t,x), T(t,x)]$. The spatial dependence of $\chi$ in inhomogeneous backgrounds was highlighted in [Tremblay, Arai, Siggia].

From a physics perspective --- given the assumption of local equilibrium --- this makes perfect sense, as it is intuitively clear what the (local) susceptibility is. It is the global susceptibility confined to a finite (coarse grained) region that must be larger than the local equilibration scale. Mathematically the double derivative $\frac{\partial^2}{\partial \mu(x,t)^2}$ may cause worries, especially when considered in the sense of distributions. This is answered in footnote 6, where we show that it is precisely the coarse graining over scales larger than the local equilibration scale that reduces the dynamical bilocal charge response --- the two point function $\langle n(x,t)n(x't')\rangle$ --- to the local charge susceptibility $\chi(\bar{x})$ with $\bar{x}$ coarse-grained.

---

## Round 2 · Author Response

We would like to thank the referees for their reports.
Regarding the first report, we have already addressed their points in our comments and indicated what changes in our re-submission were made.
As for the second report, we have expanded our conclusion to highlight the physical implication of our theory and summarized some of the key results derived in the companion paper.

---

## Round 2 · List of Changes

- Added footnote p.16 "This mimicks the charge distribution of a frozen atomic, or more appropriately ionic, lattice with valence electrons."
- Added "-- a long-time long-wavelength perturbation around thermodynamic equilibrium of a conserved charge associated to a global symmetry --" after "hydrodynamic fluctuations" on page 2.
- New footnote 4 p.3
- Added "-- which we assumed to be parity-invariant, see [12] for more general cases --" above eq 6.
- Changed sentence "This is the reference frame where the fluid velocities vanish" p.4 to "In a non-relativistic system, the static equilibrium is uniquely determined. In a relativistic system -- which we use in this paper -- it is convenient to choose the reference frame for which the fluid is at rest."
- Added "which essentially determines the low frequency long-wavelength response" in p.17
- Expanded conclusion with new paragraphs

---

## Editorial Decision

resubmitted